# The importance of digital elevation model accuracy in $\mathbf{X}_{CO_2}$ retrievals: improving the OCO-2 ACOS v11 product

Nicole Jacobs[1,2], Christopher W. O'Dell[1], Thomas E. Taylor[1], Thomas L. Logan[3], Brendan Byrne[3], Matthäus Kiel[3], Rigel Kivi[4], Pauli Heikkinen[4], Aronne Merrelli[5], Vivienne H. Payne[3], and Abhishek Chatterjee[3]

[1]Cooperative Institute for Research in the Atmosphere, Colorado State University, Fort Collins, CO, USA
[2]Department of Physics, University of Toronto, Toronto, ON, Canada
[3]Jet Propulsion Laboratory, California Institute of Technology, Pasadena, CA, USA
[4]Space and Earth Observation Centre, Finnish Meteorological Institute, Sodankylä, Finland
[5]Department of Climate and Space Sciences and Engineering, University of Michigan, Ann Arbor, MI, USA

**Correspondence:** Nicole Jacobs (n.jacobs@utoronto.ca)

**Abstract.** Knowledge of surface pressure is essential for calculating column average dry-air mole fractions of trace gases, such as $CO_2$ ($X_{CO_2}$). In the NASA Orbiting Carbon Observatory 2 (OCO-2) Atmospheric Carbon Observations from Space (ACOS) retrieval algorithm, the retrieved surface pressures have been found to have unacceptable errors, warranting a parametric bias correction. This correction depends on the difference between retrieved and a priori surface pressures, which are derived from a meteorological model that is hypsometrically adjusted to the surface elevation using a digital elevation model (DEM). As a result, the effectiveness of the OCO-2 bias correction is contingent upon the accuracy of the referenced DEM. Here, we investigate several different DEM datasets for use in the OCO-2 ACOS retrieval algorithm: the OCODEM used in ACOS v10 and previous versions, the NASADEM+ (a composite of SRTMv4, ASTER GDEMv3, GIMP, and RAMPv2 DEMs) used in ACOS v11, the Copernicus GLO-90 DEM, and two polar regional DEMs (ArcticDEM and REMA). We find that the NASADEM+ (ASTER GDEMv3) has a persistent negative bias on the order of 10 to 20 m across most regions north of 60 °N latitude, relative to all the other DEMs considered (OCODEM, ArcticDEM, and Copernicus GLO-90). Variations of 10 m in DEM elevations lead to variations in $X_{CO_2}$ of approximately 0.4 ppm, meaning that the $X_{CO_2}$ from OCO-2 ACOS v11 retrievals tend to be 0.4 to 0.8 ppm lower across regions north of 60°N than $X_{CO_2}$ from OCO-2 ACOS v10. Our analysis also suggests that the Copernicus GLO-90 has superior global continuity and accuracy compared to the other DEMs, motivating a post-processing update from OCO-2 v11 lite files (which used NASADEM+) to OCO-2 v11.1 by substituting the Copernicus GLO-90 globally. We find that OCO-2 v11.1 improves accuracy and spatial continuity in the bias-corrected $X_{CO_2}$ product relative to both v10 and v11 in high latitude regions, while resulting in marginal or no change in most regions within $\pm\ 60°$ latitude. In addition, OCO-2 v11.1 provides increased data throughput after quality control filtering in most regions, partly due to the change in DEM, but mostly due to other corrections to quality control parameters. Given large-scale differences north of 60° N between the OCODEM and NASADEM+, we find that replacing the OCODEM with NASADEM+ yields a $\sim 100$ TgC shift in inferred carbon uptake for the zones spanning 30 to 60° N and 60 to 90° N, which is on the order of 5 to 7 % of the estimated pan-Arctic land sink. Changes in inferred fluxes from replacing the OCODEM with the Copernicus GLO-90

are smaller, and given the evidence for improved accuracies from this DEM, this suggests that large changes in inferred fluxes from the NASADEM+ are likely erroneous.

# 1  Introduction

Developing a robust understanding of global carbon dynamics and predicting future climate scenarios require globally representative, highly accurate and precise observations of atmospheric greenhouse gas concentrations that cover an extended period of time (National Academies of Sciences, Engineering, and Medicine, 2018). Satellite-based spectrometers are now offering unprecedented opportunities to continuously monitor greenhouse gases on regional and global scales. The Greenhouse gas Observing Satellite (GOSAT), operating since 2009, and the NASA Orbiting Carbon Observatory 2 (OCO-2), operating since 2014, have now accumulated data records long enough to describe interannual climate variations and characterize seasonal cycles (Guan et al., 2023; Mitchell et al., 2023; Villalobos et al., 2022; Jiang et al., 2022; Jacobs et al., 2021). Column average dry-air mole fractions of atmospheric carbon dioxide ($X_{CO_2}$) must have a particularly high degree of precision and accuracy because variations on the order of tenths of a ppm must be distinguishable against a background concentration of approximately 400 ppm (e.g. Chevallier et al., 2014; Miller et al., 2007). Miller et al. (2007) report that in order to reduce uncertainty in $CO_2$ flux estimates derived from in situ networks, space-based measurements of $X_{CO_2}$ require accuracies within $\pm$ 0.2 ppm and precisions within $\pm$ 1 ppm, which equates to approximately 0.05% accuracy and 0.25% precision. Column average dry-air mole fractions of trace gases (referred to as $X_{gas}$) are representative of the total atmospheric column from the surface to space and are defined as the ratio of the total column abundance of a gas to the total column abundance of dry air. At different elevations and locations on Earth, the thickness of the atmosphere (column of dry air) varies, represented by variations in surface pressure. The abundance of trace gases will also vary in correspondence with the atmospheric thickness, hence, dividing by the column of dry air provides a concentration that should be independent of atmospheric thickness. As a result, accurate and precise calculations of $X_{CO_2}$ require both accurate knowledge of the column abundance of $CO_2$ and the column of dry air (via knowledge of surface pressure and atmospheric water vapour).

The OCO-2 instrument uses observed spectral radiances in the $O_2$A band to retrieve estimates of surface pressure necessary for retrieving $X_{CO_2}$ (see Sect. 2.1 and 3.2). These retrieved surface pressure estimates have been found to be biased relative to reanalysis estimates of surface pressure in every version of the OCO-2 ACOS retrieval algorithm (see supplemental materials Sect. S2). Previous analyses suggest that biases in retrieved surface pressures from OCO-2 are strongly correlated with biases in retrieved of $X_{CO_2}$ (Payne et al., 2022; Osterman et al., 2020; O'Dell et al., 2018, 2012). Similarly, a correlation between biases in retrieved surface pressure and biases in retrieved $X_{CO_2}$ was found for GOSAT by Wunch et al. (2011a). Ultimately, the OCO-2 team found that retrieving surface pressure and applying an empirical bias correction after the fact yields a more accurate $X_{CO_2}$ product. Many of the retrieval algorithms applied to real or simulated observations from GOSAT, GOSAT-2, GOSAT-GW, OCO-2, OCO-3, and CO2M either do not retrieve surface pressure and use values taken directly from a meteoro-

logical model or retrieve surface pressure and then bias correct for corresponding inaccuracies during post-processing (Someya et al., 2023; Noël et al., 2021; O'Dell et al., 2018; Reuter et al., 2017; Yoshida et al., 2013; Cogan et al., 2012; Butz et al., 2011). The OCO-2 bias correction includes a parameter referred to as $dP_{frac}$ that represents the shift in $X_{CO_2}$ that directly results from the bias in retrieved surface pressure relative to the a priori surface pressure estimate (see Sect. 3.2 and Kiel et al., 2019).

The a priori surface pressures in the OCO-2 Atmospheric Carbon Observations from Space (ACOS) retrieval algorithm are taken from the GEOS-FPIT meteorological model (Lucchesi, 2015), hypsometrically adjusted to the surface elevation, which is itself taken from a digital elevation model (DEM) (Kiel et al., 2019; O'Dell et al., 2018; Osterman et al., 2020). Correcting with respect to $dP_{frac}$ (defined by Kiel et al. (2019)) has the effect of making the bias corrected $X_{CO_2}$ more dependent on the a priori surface pressure estimate and less dependent on the retrieved surface pressure. Consequently, inaccuracies in the

DEM can propagate into the a priori surface pressure and yield inaccuracies in $X_{CO_2}$. A study by Hachmeister et al. (2022) also demonstrated the importance of the DEM in TROPOMI retrievals of $X_{CH_4}$. In their analysis, Hachmeister et al. (2022) reprocessed TROPOMI retrievals of $X_{CH_4}$ using recent ICESat-2 elevation data and found that this eliminated anomalies in $X_{CH_4}$ as large as 100 ppb along the Greenland coastline. The study by Hachmeister et al. (2022) ultimately prompted the use of the Copernicus GLO-90 DEM in the most recent update to the TROPOMI retrieval algorithm for $X_{CH_4}$ (Schneising et al.,

2023). In fact, the use of DEMs as a tool for developing more uniform gridded maps of surface pressure estimates is ubiquitous within the practice of retrieving $X_{gas}$ quantities from satellite-based observations. As a result, the importance of a globally continuous, accurate and high-spatial-resolution DEM extends to other trace gases, although the precision requirements for $X_{CO_2}$ and $X_{CH_4}$ make the impacts of DEM inaccuracies far more significant.

A DEM represents the interpretation and conversion of the real boundaries of Earth's spheres (i.e., lithosphere, biosphere, cryosphere, etc.) into a mathematical framework that can be used in practical applications. For reference, Guth et al. (2021) provide an encyclopedic glossary of DEM terminology as well as thorough explanations of structures within and uses for DEMs. For the purposes of this analysis, it is important to distinguish between digital terrain models and digital surface models (or surface terrain models) as both are often referred to as DEMs. In general, terrain models attempt to estimate the elevation of

the lithospheric boundary, excluding the heights of objects that might be considered part of another sphere, such as vegetation, architecture, or ice, while surface models estimate the elevation of the boundary between the atmosphere and the combined surface of all other non-gaseous spheres (i.e., the elevations at the tops of forest canopies, buildings, ice flows, etc.). All of the DEMs considered for inclusion in the OCO-2 ACOS retrieval algorithm and used in this analysis are surface models, and this is appropriate in the context of a trace gas retrieval because light is reflected off the surface of objects on the ground. Raw

elevation measurements used in the construction of radar-based DEMs (like Copernicus and SRTM discussed in Sect. 2.3, 2.4, and 2.5) consist of active remote sensing in which radar waves are emitted from a device on board of an aircraft, a space shuttle, or a satellite and reflected back to a detector, then the time delay or frequency shift is used to infer the distance traveled by the emitted radiation. There are potential complications that can arise in determining where the surface boundary actually resides, and there can be differences in how deep different wavelengths of light may penetrate a surface, such as a forest canopy, before

being reflected back to a detector used in either measurements of surface elevations or trace gas concentrations. The finer de-

tails of these discrepancies are largely outside the scope of this paper, and we point the reader to Guth and Geoffroy (2021), Qi and Dubayah (2016), and Sexton et al. (2009). It is also worth noting that each DEM utilizes varying techniques for gridding, smoothing, and void filling in order to construct a full, continuous global map of surface elevations. Sections 2.3 through 2.7 provide some details for each DEM studied in this research.


During the development of the most recent update to the OCO-2 ACOS retrieval algorithm, version 11 (v11), the referenced DEM was updated for the first time since the inception of the OCO ACOS algorithm in 2009 (O'Dell et al., 2012; Zong, 2008). This new DEM has large differences across the Arctic north of 60° N relative to the DEM used in previous versions of ACOS, inducing large changes to the retrieved v11 $X_{CO_2}$ at high latitudes. This prompted an investigation of the DEM used in v10 and previous versions of the OCO-2 ACOS algorithm (referred to here as OCODEM, described in Sect. 2.3), the DEM used in v11 (NASADEM+, Sect. 2.4), and other recently developed DEMs including the Copernicus global DEM (Sect. 2.5), ArcticDEM (Sect. 2.6), and Reference Elevation Model of Antarctica (REMA; Sect. 2.7). It soon became clear that a robust understanding of the accuracy of these DEMs and the impact that different DEMs may have on $X_{CO_2}$ retrievals was warranted. This analysis is particularly relevant to improving OCO-2 retrievals over high latitude terrestrial regions because these regions have the largest discrepancies amongst DEMs, and have historically been excluded from high-spatial-resolution DEMs or received less attention in assessments of DEM quality (Karlson et al., 2021; Noh and Howat, 2015; Cook et al., 2012).

Northern high latitude regions are experiencing climate change at an increased rate relative to other regions as a result of polar amplification (Smith et al., 2019; Park et al., 2018; Pithan and Mauritsen, 2014; Holland and Bitz, 2003; Manabe and Wetherald, 1975), yet a shortage of observations over these regions remain a significant impediment to characterizing and quantifying global carbon uptake (Byrne et al., 2020; Euskirchen et al., 2017; Barlow et al., 2015; Pan et al., 2011). This confluence of rapid change and a shortage of observations motives concerted efforts to increase and improve measurements of atmospheric $CO_2$ concentrations over the northern high latitude regions. This is complicated by the fact that a number of challenges persist to retrieving column concentrations of $CO_2$ over high latitude regions. Most notably, high solar zenith angles that correspond to high airmass in the slant column path of radiation that continue even in the summer season, as well as a near or complete absence of sun light that prevents passive remote sensing during polar winter. The higher airmasses result in larger aerosol optical depths and smaller scattering angles, which increase the negative impacts of aerosol scattering. Snow and ice covered surfaces also present a challenge to many methods of remote sensing, especially using infrared wavelengths that tend to have low reflectivity over these types of surfaces. Until recently, there were many sources of uncertainty in high latitude OCO-2 retrievals, and it is only with targeted efforts to improve our understanding of high latitude retrievals over several versions of the ACOS OCO-2 algorithm (Mendonca et al., 2021; Jacobs et al., 2020) that the impact of inaccuracies in the DEM have been recognized. Simultaneously, many challenges have also hindered the development, mapping, and validation of DEMs over high latitude regions, specifically due to the abundance of low-contrast and repetitively patterned surfaces associated with snow and ice, as well as topographical discontinuities associated with cliff and ice shelf faces (Karlson et al., 2021; Noh and Howat, 2015; Cook et al., 2012). Only in the past five to ten years have new methods for data collection with

satellites and statistical treatment of observations allowed for substantial improvement in the accuracy and quality of DEMs over high latitudes (Fahrland et al., 2020; Marešová et al., 2022; Karlson et al., 2021; Noh and Howat, 2015). Several DEMs released in the last few years have pan-Arctic coverage and include updates to their treatment of northern high latitudes, including ASTER GDEMv3 (Gesch et al., 2016) in 2016, ALOS World 3D (Takaku et al., 2020) in 2016, the Copernicus GLO-90 in 2020 (Fahrland et al., 2020), and ArcticDEM in 2018 (Porter et al., 2018).

In this paper we compare mapped elevations across several DEMs, including the OCODEM, NASADEM+, Copernicus GLO-90, ArcticDEM, and REMA, leading to the identification of significant quality improvements in the more recently available DEMs relative to the older OCODEM and NASADEM+ elevation collections. The corresponding effects of the choice of DEM on the bias-corrected $X_{CO_2}$ retrievals from OCO-2 and inferred $CO_2$ fluxes are then explored. Overall, we endeavour to inform OCO-2 data users on recent changes and improvements to OCO-2 ACOS retrievals in the most recent v11.1 update as a result of these findings. We also seek to highlight the importance of using an accurate DEM not only for OCO-2, but for any space-based retrievals of trace gases, especially $X_{CO_2}$ and $X_{CH_4}$ due to the high precision and accuracy requirements of these gases.

## 2    Datasets

### 2.1    OCO-2 observations and retrievals

The NASA Orbiting Carbon Observatory 2 (OCO-2) is a passive polar-orbiting satellite launched in 2014 that began collecting data in September 2014 (Eldering et al., 2017). It detects radiances in three spectral bands at 0.765 $\mu$m (O$_2$A band), 1.61 $\mu$m (weak $CO_2$ band), and 2.06 $\mu$m (strong $CO_2$ band). There are three viewing modes, nadir, glint, and target. Nadir observations are taken with the instrument pointed straight down, roughly normal to the ground below the instrument. Glint observations are taken with the instrument aligned such that the viewing angle is equal to the angle of reflection of the incident sunlight. For target mode observations the OCO-2 instrument scans back and forth collecting as many soundings as possible covering a $0.46° \times 0.8°$ box around a specified location. Unlike, OCO-3, OCO-2 has a relatively limited capacity for number of target sites that can be preprogrammed with the majority of these sites in the Total Column Carbon Observing Network (TCCON; see Sect. 2.2). During normal operations (nadir or glint mode) OCO-2 observes 8 adjacent soundings, referred to as footprints, that span the narrow swath (< 10 km) of the instrument field of view. OCO-2 footprints have dimensions of approximately 1.3 $\times$ 2.25 km in nadir mode (Crisp et al., 2008), and these dimensions can vary on the order of $\pm 1$ km with other viewing geometries. It collects 24 soundings per second, yielding approximately 5.5 million soundings each month that pass preliminary cloud screening and are included in the OCO-2 lite files (Crisp et al., 2021, 2017). Crisp et al. (2021) report single-sounding precision in $X_{CO_2}$ of approximately 0.5 ppm and accuracy within 1 ppm. The ACOS algorithm is used to retrieve $X_{CO_2}$ from OCO-2 observed radiances. For more details on OCO-2 instrument operations, one may reference the Algorithm Theoretical Basis Document (Crisp et al., 2021).

**Table 1.** DEMs used in each version of the OCO-2 ACOS algorithm included in this analysis.

| OCO-2 ACOS version | DEM | Section on DEM |
|---|---|---|
| v10 | OCODEM | 2.3 |
| v11 | NASADEM+ | 2.4 |
| v11.1 | Copernicus DEM | 2.5 |

The ACOS algorithm was first developed and used for GOSAT observations beginning in 2009 and was later modified for use with OCO-2 (O'Dell et al., 2018, 2012). OCO-2 ACOS v10 was released in 2020 (Taylor et al., 2023; Osterman et al., 2020) and v11 was released in 2023 (Payne et al., 2022). More details of other changes made during the update from OCO-2 ACOS v10 to v11 are discussed in Appendix A. Based on results discussed in this paper, the decision was made to release the v11.1 update (discussed in Sect. 3.2), which implements the Copernicus GLO-90 in place of the NASADEM+ with a recalculated bias correction to account for this change. In addition, v11.1 includes modifications to the quality control parameters h2o_ratio and co2_ratio (ratio of retrieved $H_2O$ in the weak $CO_2$ band to retrieved $H_2O$ in the strong $CO_2$ band and ratio of retrieved $CO_2$ in the weak $CO_2$ band to retrieved $CO_2$ in the strong $CO_2$ band, respectively; see details in Appendix B) to make them more accurate. Although both OCO-2 v11 and v11.1 retrievals are publicly available, we encourage the use of v11.1 as a superior data product. Changes in the fraction of OCO-2 retrievals that pass quality controls as a result of the change in DEM or modifications to h2o_ratio and co2_ratio are shown if Fig. 8 and discussed in Sect. 4.4. The OCO-2 v11.1 update is only applied to retrievals over land, so soundings over ocean are unchanged from v11. Table 1 summarizes the DEMs used in each version of the OCO-2 ACOS algorithm included in this analysis.

A global bias correction is applied to all OCO-2 retrievals of $X_{CO_2}$ that corrects for systematic biases from several parameters in the retrieval including, most notably for this analysis, surface pressure bias (eg., $dP_{frac} = X_{CO_2,raw}(1 - P_{ap,\ sco2}/P_{ret})$, see Sect. 3.2) (Payne et al., 2022; Osterman et al., 2020; Kiel et al., 2019). The bias correction also includes a multiplicative scaling based on comparisons to ground-based measurements from the Total Carbon Column Observing Network (TCCON) and a footprint bias correction, described in more detail in Payne et al. (2022) and Osterman et al. (2020).

## 2.2 TCCON data

The Total Carbon Column Observing Network (TCCON) is a global network of ground-based, high-resolution, solar-viewing spectrometers using the Bruker IFS-125HR instrument (Wunch et al., 2015, 2011a), and is the primary source of ground-based validation for OCO-2 and OCO-3 (Payne et al., 2022; Osterman et al., 2020; Wunch et al., 2017). A multitude of studies have used TCCON data as a source of validation data for satellite-based measurements of greenhouse gases including Jalali et al. (2022), Lorente et al. (2021), Yang et al. (2020), Hedelius et al. (2019), Kulawik et al. (2016), and Wunch et al. (2011b), as just a few examples. Furthermore, the European Space Agency has made TCCON data the official ground-truth for TROPOMI and

upcoming satellite missions. In Sect. 4.5, we use TCCON observations to assess how biases in OCO-2 observations change across ACOS v10, v11, and v11.1 retrievals and explore how these changes in bias may be attributed to differences in the DEMs. While OCO-2 biases are evaluated over most TCCON sites (see list of sites in Table 4), significant differences among OCO-2 ACOS versions at the TCCON site in Sodankylä, Finland (67.37° N, 26.63° E; Kivi et al., 2022; Kivi and Heikkinen, 2016) are reviewed in more detail. Sodankylä is chosen for further study because it is one of the two northern high latitude

sites that exhibit relatively large changes in OCO-2 bias as a result of changing the DEM in OCO-2 ACOS versions v10, v11, and v11.1, as well as offering a large number of OCO-2 overpasses with corresponding ground-based measurements.

## 2.3 OCODEM (DEM for v10 and previous OCO ACOS versions)

Before the development of ACOS v11, the referenced DEM had not been changed or updated since the inception of the OCO
ACOS algorithm. This DEM, which we refer to as OCODEM, is composed of data from 1) the 2000 Shuttle Radar Topography Mission (SRTMv1; Farr et al., 2007) within $\pm$ 60° latitudes; 2) a mixture of DTED level 1 (90 m resolution) and GTOPO30 (1 km resolution vector source) for regions north of 60° latitude; 3) the Radarsat Antarctic Mapping Project version 2 (RAMPv2; Liu et al., 2015, ; 200 m resolution) DEM collection (see additional details in Sect. 2.4) for Antarctica. Zong (2008) reports the desired OCO vertical accuracy goal of $\pm$ 12.5 m was only achieved by the SRTM data, as there were no other equivalent
alternatives available for the other areas at the time (in 2007).

Due to access restrictions on the OCODEM, altitudes had to be extracted from OCO-2 soundings and regridded to $0.1° \times 0.1°$ grid. Reported sounding altitudes and location coordinates for each OCO-2 observation (defined as described in Sect. 3.1) are used to determine which OCO-2 v10 soundings fall within a given tenth degree grid cell and then averages of all sounding
altitudes in each grid cell are used to reconstruct an approximation of the source DEM in OCO-2 v10 (the OCODEM). While this method is somewhat convoluted, it should be reasonably accurate provided that there are sufficient numbers of OCO-2 soundings and the coverage is spatially consistent over the entire globe. However, OCO-2 coverage is not perfectly continuous or even spatially consistent across different regions. As shown in Sect. S1 of the supplemental materials, there are smaller numbers of OCO-2 soundings that tend to follow a pattern along the orbital path of the instrument, and this is especially
prominent over the Southern mid-latitudes and the tropics. As a result, it is likely that some of the striated patterns of elevation differences on the order of 10 to 15 m seen (mostly in the southern hemisphere) in Fig. 1 panels (a) and (b) are an artifact of this aggregation process and not real differences between the OCODEM and the other DEMs.

## 2.4 NASADEM+ (DEM for OCO-2 v11)

The "NASADEM+" database collection was assembled by the Jet Propulsion Laboratory (JPL) in 2019 for general NASA Mission support and was not prepared specifically for OCO-2. It is composed of data from five distinct DEMs. The "NASADEM" is used in the NASADEM+ for all regions within $\pm$ 60° latitude, and is composed of data from the Shuttle Radar Terrain Model

version 4 (SRTMv4). SRTMv4 uses data collected in the original SRTM mission in 2000 with a number of improvements including height calibrations from ICESat, void filling, and an improved water mask (Crippen et al., 2016; Simard et al., 2016).

Simard et al. (2016) show that the NASADEM no longer has the systematic biases relative to ICESat/GLAS data that were observed across the contiguous United States in SRTM version 3 and they report an RMSE of 2.3 m over this region. The Advanced Spaceborne Thermal Emission and Reflectance Radiometer Global DEM version 3 (ASTER GDEMv3; Abrams et al., 2020) is used for 60° N to 83° N, excluding Greenland, as well as for the Palmer Peninsula in Antarctica bounded by 63° to 69° S and 57° to 68° W. The ASTER GDEMv3 was chosen for the high latitude regions because of its completeness of global

coverage, free public license, and the lack of comparable alternatives. Later analysis over a site in Sweden by Karlson et al. (2021) found significant terrain differences between ASTER GDEMv3 and the Copernicus GLO-90, ALOS, and ArcticDEM collections, as well as the Swedish National DEM they used as reference. In their evaluation of SRTM combined with the older ASTER GDEMv2, Tighe and Chamberlain (2009) found terrain errors of 15 to 19 m RMSE, which is consistent with findings by Tachikawa et al. (2011) who report errors in ASTER GDEMv2, at 17 m RMSE. Gesch et al. (2016) found that ASTER

GDEMv3 errors were improved relative to ASTER GDEMv2 over the coterminous U.S., but they did no validation over high latitudes. The Greenland Ice Sheet Mapping Project (GIMP) is used in the NASADEM+ for elevations over Greenland. Howat et al. (2014) reports errors in GIMP elevations of ±10 m RMSE, though most ice surfaces are accurate to ±1 m and high relief areas can have errors upwards of ±30 m. The Advanced Land Observing Satellite version 3.1 (ALOSv3.1; Takaku et al., 2020), developed by JAXA, is used in the NASADEM+ for 83° N to 84° N. Finally, for Antarctica (61° to 90° S, excluding

the Palmer Peninsula region filled in by ASTER GDEMv3) the NASADEM+ uses data from the RAMPv2, which is the same data used in the previous OCODEM. RAMPv2 is based on radar observations from 1949 through 1999 and Liu et al. (1999) approximates height errors to be in the range of 1 to 100 m depending on location. Though some improvements have been made in the RAMP update to version 2, as described by Liu and Jezek (2004), these changes are isolated to a few study areas along the coast of Antarctica for which direct validation observations exist. Most of the source datasets for the NASADEM+

collection have spatial resolutions of 1 arcseconds or approximately 30 m, but the RAMPv2 data has spatial resolution of approximately 200 m. The combined product used in the OCO-2 retrievals is scaled to 3 arcseconds or approximately 90 m resolution.

## 2.5 Copernicus GLO-90 (DEM for OCO-2 v11.1)

As with the OCODEM and NASADEM+, the Copernicus global DEM is a surface terrain model that includes the heights of infrastructure and vegetation (Fahrland et al., 2020). It is derived from the WorldDEM with some changes and improvements to inland waters, coastlines, and other small scale features. The WorldDEM is based on data from the TanDEM-X Mission and managed by a public-private partnership between the German Aerospace Centre (DLR) and Airbus Defence and Space. The WorldDEM has historically had restricted access, but with the release of the Copernicus global DEM to the general

public in autumn of 2021, the advantages of the TanDEM-X data can be more broadly utilized by the international scientific community. Marešová et al. (2022) report that for especially rough terrain in three European mountain ranges improvements

in the Copernicus global DEM yield an average reduction in RMSE from 28 m for raw TanDEM-X data to 9 m. Fahrland et al. (2020) report an absolute and relative global mean vertical accuracy no worse than 4 m, though they concede that local variations in accuracy may be larger. These reported metrics exclude Greenland and Antarctica due to complications in validation analyses over regions with permanent ice and snow; however, we show in Fig. 2 that the Copernicus DEM is in good agreement with the ArcticDEM (see Sect. 2.6) over Greenland and we show in Fig. 3 that the Copernicus DEM is in good agreement with REMA (see Sect. 2.7) over Antarctica. Both the ArcticDEM and REMA are validated using ICESat-2, which is also shown by Hachmeister et al. (2022) to improve $X_{CH_4}$ retrievals from TROPOMI over Greenland. The findings of Hachmeister et al. (2022) prompted a change to the Copernicus GLO-90 DEM in the most recent update to TROPOMI $X_{CH_4}$ retrievals. As a result, Schneising et al. (2023) report reduced errors in assumed surface pressure and retrieved $X_{CH_4}$ on the order of 1%, with notable improvements over high latitude regions. Karlson et al. (2021) show that the Copernicus global DEM has the best vertical accuracy in terms of mean error, standard deviation, and RMSE over their study areas in Sweden, when compared to ASTER GDEMv3, ALOS, and ArcticDEM. A number of other studies found that the Copernicus global DEM performs as well or better than other DEMs in lower latitude regions as well (e.g. Li et al., 2022; Carrera-Hernández, 2021; Guth and Geoffroy, 2021). In a very thorough global comparison of available DEMs, Bielski et al. (2023) found the overall robustness, accuracy, and precision of the Copernicus global DEM to be better than other DEMs. Of the top three benchmarked DEMs in the study by Bielski et al. (2023), the Copernicus global DEM is the only publicly accessible DEM that meets the requirement of the OCO-2 ACOS retrieval algorithm for a globally consistent and void-free surface terrain model.

The Copernicus global DEM has been produced as 30 m (∼1 arcseconds) and 90 m (∼3 arcseconds) resolution gridded products, referred to as GLO-30 and GLO-90, respectively. In this analysis and in the ACOS OCO-2 v11.1 update, the Copernicus GLO-90 is used. This matches the resolution of the OCODEM and NASADEM+ products that are also considered in this study.

## 2.6  ArcticDEM

The ArcticDEM was created as a NGA-NSF public-private initiative using the WorldView satellite constellation (Porter et al., 2018). It covers regions north of 60° N, with some voids dispersed throughout these regions, as well as some coverage that extends south of 60°N. The mosaic tile product is originally developed at 2 m resolution with lower resolution products available up to 1 km; in this analysis the 32 m mosaic tile product is used. The ArcticDEM is an automated stereo-photogrammetric digital surface model generated using optical imagery from the WorldView satellites and a technique called Surface Extraction with TIN-based Search-space Minimization (SETSM), where TIN is an acronym of Triangular Irregular Network (Noh and Howat, 2015). Noh and Howat (2015) demonstrate that SETSM provides improved accuracy over low-contrast and repetitively textured terrain, such as the snow and ice covered regions at high latitudes. After vertical registration to Cryosat-2 and ICESat altimetry, Porter et al. (2018) claim absolute uncertainties of less than 1 m over most of the covered regions. Karlson et al. (2021) report that the ArcticDEM performed second best overall, after the Copernicus GLO-90, over their study areas in Sweden. It also has the highest available spatial resolution of the DEMs tested while maintaining vertical accuracy on par with the Copernicus GLO-90, though Karlson et al. (2021) note that this can be offset by the fact that the ArcticDEM has more voids

and trouble mapping water bodies.

## 2.7 REMA

As part of the same initiative as the ArcticDEM, the Reference Elevation Model of Antarctica (REMA) is a high resolution
DEM covering Antarctica (Howat et al., 2022). The REMA is also developed using data from the WorldView satellite constellation with vertical registration using Cryosat-2 and ICESat altimetry, and is available as mosaic tile products from 2 m to 1 km resolution. Here, the 32 m mosaic tile product is used.

# 3 Methods

## 3.1 Treatment and aggregation of DEM data

For the purpose of simplifying plots and matching DEMs that are usually not mapped to exactly matching coordinates, DEMs were upscaled to $0.01°$, $0.1°$ or $0.5°$ resolution by taking the average of all DEM data points that lie within a given aggregation grid cell. The specific spatial resolutions are given in the figure captions for relevant figures. In addition, all DEMs are adjusted to the EGM96 (Earth Gravitational Model 1996) geoid.

OCO-2 retrievals have a reported sounding latitude and longitude coordinate that is approximately the center of the sounding footprint ($\sim 1.3 \times 2.25$ km), and the boundaries of this sounding footprint are defined by an average of the vertex coordinates in the three bands. Within the ACOS algorithm, altitudes for specific OCO-2 soundings are calculated as the average of DEM data points that fall within the boundaries of the sounding footprint. The new lite file update to OCO-2 v11, labelled v11.1, uses the Copernicus global DEM at 3 arcseconds resolution ($\sim 90$ m; aka Copernicus GLO-90) in place of the NASADEM+ for all soundings, globally. The v11.1 update is fundamentally a change to sounding altitude, as well as a number of other retrieval parameters that depend on the sounding altitude, applied as a post-processing correction to the v11 retrievals. The ACOS Level 2 Full Physics (L2FP) retrieval, which is computationally expensive, was not rerun for the v11.1 update.

## 3.2 The role of the DEM in OCO-2 retrievals

Atmospheric column average dry air mole fractions of $CO_2$ ($X_{CO_2}$) are defined as

$$X_{CO_2} = \frac{\text{column } CO_2}{\text{column dry air}}. \tag{1}$$

where each of the column terms have typical units of mol m$^{-2}$. For OCO-2 retrievals, the column of dry air is determined as a function of the retrieved surface pressure $P_{ret}$ and the atmospheric profile of specific humidity $q$, as follows:

$$\text{column dry air} = \int_0^{P_{ret}} \frac{(1-q)\,dP}{g\,M_{dry}} \tag{2}$$

where $M_{dry}$ is the molar mass of dry air ($\sim 28.96 \cdot 10^{-3}$ kg/mol) and $g$ is the local acceleration due to gravity, which has a slight dependence on latitude and altitude. Using the fact that the total column of water vapour (TCWV, typically expressed kg m$^{-2}$) equals $\int_0^P \frac{q}{g} dP$, Eq. 2 can be simplified to

$$\text{column dry air} \cong \frac{1}{M_{dry}}\left(\frac{P_{ret}}{\bar{g}} - \text{TCWV}\right) \tag{3}$$

The surface pressure term in Eq. 3 is typically more than 100 times larger than the TCWV term, implying that the column of dry air is proportional to the surface pressure. Errors in either retrieved surface pressure $P$ or retrieved TCWV can lead to errors in $X_{CO_2}$. Fortunately, errors in OCO-retrieved TCWV are $\sim 1$ kg m$^{-2}$ (Nelson et al., 2016); against a typical surface pressure of $10^5$ Pa, this yields a 0.01% error in the dry air column, and hence a 0.01% in $X_{CO_2}$ ($\sim$0.04 ppm). However, retrievals of surface pressure from ACOS using GOSAT or OCO-2 spectra are poor, with RMS errors on the order of 3 hPa, which yield $X_{CO_2}$ errors on the order of 1 ppm. In contrast, most modern reanalyses are believed to be better than 1 hPa. We find empirically that the biases in retrieved $X_{CO_2}$ (relative to TCCON observations or model estimates) are strongly correlated with the difference between retrieved and a priori surface pressure (dP) (Wunch et al., 2011b; O'Dell et al., 2012, 2018; Osterman et al., 2020). This correlation motivates the inclusion of a parametric correction with respect to the term dP$_{frac}$, defined as follows (Kiel et al., 2019):

$$\text{dP}_{frac} = X_{CO_2,raw}(1 - P_{ap,\ sco2}/P_{ret}), \tag{4}$$

where $X_{CO_2,raw}$ is the retrieved $X_{CO_2}$ before any bias correction, $P_{ap,\ sco2}$ is the a priori surface pressure in the strong CO$_2$ band and $P_{ret}$ is the retrieved surface pressure from the full physics retrieval. The OCO-2 ACOS v10 bias correction for soundings over land is

$$X_{CO_2} = \frac{X_{CO_2,\ raw} - \text{Feats} - \text{footprint\_bias}}{\text{divisor}} \tag{5}$$

where the divisor is based on a global offset relative to TCCON and

$$\text{Feats} = -0.855(\text{dP}_{frac}) + (\text{other parameters}). \tag{6}$$

Osterman et al. (2020) and Payne et al. (2022) provide full definitions of the global bias corrections in v10 and v11, respectively. Most of the sensitivity to the DEM in OCO-2 retrievals is due to the parametric correction as a function of dP$_{frac}$ in the global bias correction.

**Table 2.** Variables that have changed values in OCO-2 v11.1 relative to v11, due to the substitution of the Copernicus GLO-90.

| Data group and name in lite files | Definition |
| --- | --- |
| Meteorology/psurf_apriori_o2a,wco2,sco2 | a priori surface pressure in each individual OCO-2 band ($O_2A$, weak $CO_2$, and strong $CO_2$) |
| Preprocessors/dp_abp | retrieved minus a priori surface pressure in the A-band preprocessor |
| Sounding/altitude | average DEM elevation within the sounding footprint |
| Sounding/altitude_stddev | standard deviation of DEM elevations within the sounding footprint |
| Retrieval/dp_o2a,sco2 | retrieved minus a priori surface pressure individually in the $O_2A$ and strong $CO_2$ band |
| Retrieval/dpfrac | see Eq. 4 and description in Kiel et al. (2019) |
| xco2 | retrieved $X_{CO_2}$ with global bias correction applied |
| xco2_quality_flag | binary indicator of passing quality control filters |
| xco2_qf_bitflag | a 32-bit integer indicating the pass=0 or fail=1 status of each of the quality control filters |
| xco2_qf_simple_bitflag | a 8-bit indicating the pass or fail status of groups of quality control filters, as defined in Payne et al. (2022), Osterman et al. (2020) |

**Table 3.** New variables introduced in OCO-2 v11.1 lite files that do not exist in v11.

| Data group and name in lite files | Definition |
| --- | --- |
| Auxiliary/xco2_quality_flag_b11_original | original quality control binary indicator from B11 by sounding ID |
| Auxiliary/altitude | original v11 sounding altitude |
| Auxiliary/altitude_stddev | original v11 altitude_stddev |
| Auxiliary/tvirtual | virtual temperature used in adjusting the a priori surface pressures via the hypsometric equation |
| co2_ratio_bc | bias-corrected version of co2_ratio (see Appendix B) |
| h2o_ratio_bc | bias-corrected version of h2o_ratio (see Appendix B) |

The v11.1 lite file update involved changing sounding altitudes to reference the Copernicus GLO-90 as a post-processing correction. Changes were also made to retrieval parameters that directly depend on the sounding altitude including a priori surface pressure, the bias corrected $X_{CO_2}$ product (xco2), and other parameters as listed in Table 2. Some new parameters, as listed in Table 3, were added to the OCO-2 v11.1 lite files that did not exist in v11 or previous versions of the OCO-2 ACOS 345 algorithm. The OCO-2 v11 L2Std product remains unchanged and only soundings over land are affected.

### 3.3 Data handling for TCCON comparisons

When comparing OCO-2 and TCCON retrievals, OCO-2 soundings are considered coincident to TCCON if they are within $2.5°$ latitude and $2.5°$ longitude of the location of the TCCON site, and are compared against an average of any TCCON measurements collected within $\pm$ 1 hour of the average OCO-2 overpass time for that site. Only OCO-2 overpasses that have at least 100 good quality soundings and at least 10 TCCON measurements within the 2 hour period around the mean overpass time are included. TCCON data used in this analysis were processed with the GGG2020 retrieval algorithm and have been corrected for discrepancies with OCO-2 retrievals in the a priori $CO_2$ profile and averaging kernel using methods described by Mendonca et al. (2021). The TCCON sites included are listed in Table 4 along with the time period of TCCON data available for inclusion in this analysis. When differences between retrieval parameters in different OCO-2 ACOS versions are discussed in this paper, as well as with the comparisons to TCCON measurements, OCO-2 soundings are paired as the intersection of quality control filters across versions. This means that only soundings that pass quality controls in all versions of the OCO-2 ACOS retrieval algorithm are included in these comparisons.

### 3.4 Evaluating the impact of the DEM on $CO_2$ flux estimates

To isolate the impact of changing the DEM on inferred $CO_2$ fluxes, we performed flux inversion analyses using v10 retrievals with a priori surface pressures hypsometrically adjusted for a change in altitude. This approach excludes the effects of other changes to the retrieval algorithm from the v11 update. The adjusted altitudes are taken directly from the v11 and v11.1 lite files by matching soundings to ensure the integrity of the footprint mapping methods used in the official retrievals. In effect, this change occurs only through modifying the value of $dP_{frac}$ in the bias correction, and all other retrieval parameters, bias correction coefficients, and global scaling factor remain unchanged from the official v10 release. Each set of assimilated $X_{CO_2}$ data is first aggregated to "super observations" by averaging fields over a $0.5° \times 0.625°$ spatial grid, similar to what is done in the OCO-2 Model Intercomparison Project (OCO2-MIP Peiro et al., 2022).

Three sets of atmospheric $CO_2$ inverse analyses were conducted for each of the original v10 (with OCODEM), NASADEM+ modified, and Copernicus GLO-90 modified $X_{CO_2}$ Land Nadir + Land Glint (LNLG) datasets. These inversions follow the set-up of Byrne et al. (2020), and employ the CMS-Flux system with tracer transport at $4° \times 5°$ degree spatial resolution using MERRA-2 reanalysis fields. We optimize 14-day scaling factors for each $4° \times 5°$ degree grid cell on net ecosystem exchange (NEE) and ocean surface-atmosphere fluxes for October 2017 through March 2019. This is performed using three different prior NEE datasets, which are described in Byrne et al. (2020). As a result, a mini-ensemble of flux estimates is generated for each of the three $X_{CO_2}$ datasets, yielding a total of nine model runs. Posterior fluxes are examined for 2018 only. See Sect. 3 of Byrne et al. (2020) for additional details on the inversion set-up.

**Table 4.** References to all TCCON sites used in this analysis. Time spans indicate the period of TCCON data included in the analysis, but do not necessarily represent the time period for which OCO-2 overpasses that pair with TCCON measurements exist.

| Site | Location | Time span | Reference |
|---|---|---|---|
| Lauder | 45.04° S, 169.68° E | 2014-09 to 2022-06 | Sherlock et al. (2022) and Pollard et al. (2022) |
| Burgos | 18.53° N, 120.65° E | 2017-03 to 2021-04 | Morino et al. (2022c) |
| Izana | 28.31° N, 16.5° W | 2014-09 to 2022-10 | García et al. (2022) |
| Hefei | 31.9° N, 119.17° E | 2015-11 to 2020-12 | Liu et al. (2022) |
| Saga | 33.24° N, 130.29° E | 2014-09 to 2022-08 | Shiomi et al. (2022) |
| Pasadena | 34.14° N, 118.13° W | 2014-09 to 2022-10 | Wennberg et al. (2022a) |
| Edwards | 34.96° N, 117.88° W | 2014-09 to 2022-10 | Iraci et al. (2022) |
| Nicosia | 35.14° N, 33.38° E | 2019-09 to 2021-06 | Petri et al. (2022) |
| Xianghe | 39.8° N, 116.96° E | 2018-06 to 2022-02 | Zhou et al. (2022) |
| Tsukuba | 36.05° N, 140.12° E | 2014-09 to 2021-03 | Morino et al. (2022b) |
| Lamont | 36.6° N, 97.49° W | 2014-09 to 2022-10 | Wennberg et al. (2022c) |
| Rikubetsu | 43.46° N, 143.77° E | 2014-09 to 2021-06 | Morino et al. (2022a) |
| Park Falls | 45.94° N, 90.27 W | 2014-09 to 2022-10 | Wennberg et al. (2022b) |
| Garmisch | 47.48° N, 11.06° E | 2014-09 to 2021-10 | Sussmann and Rettinger (2023) |
| Orleans | 47.96° N, 2.11° E | 2014-09 to 2021-07 | Warneke et al. (2022) |
| Paris | 48.85° N, 2.36° E | 2014-09 to 2022-03 | Té et al. (2022) |
| Karlsruhe | 49.1° N, 8.44° E | 2014-09 to 2021-12 | Hase et al. (2022) |
| Bremen | 53.1° N, 8.85° E | 2014-09 to 2021-06 | Notholt et al. (2022) |
| East Trout Lake | 54.35° N, 104.99° W | 2016-10 to 2022-08 | Wunch et al. (2018) |
| Sodankylä | 67.37° N, 26.63° E | 2014-09 to 2022-06 | Kivi et al. (2022) |
| Eureka | 80.05° N, 86.42° W | 2014-09 to 2020-07 | Strong et al. (2022) |

## 4 Results

### 4.1 Direct comparisons amongst DEMs

The largest differences among DEMs are concentrated in the polar regions (i.e., the northern high latitudes north of 60° N and Antarctica). For the OCODEM, this could be expected given the age and low resolutions of the DTED/GTOPO30 mixture used in the higher northern latitudes and the RAMPv2 DEM used in Antarctica. The RAMPv2 DEM data (Sect. 2.4) was carried forward from the OCODEM to the NASADEM+ collection (due to lack of an update proponent at the time), passing along its low resolution and accuracy issues. The discrepancies between the NASADEM+ and the Copernicus GLO-90 over the regions north of 60° N (ASTER GDEMv3) and Antarctica (RAMPv2) are clearly visible in Fig. 1, with more focused comparisons in

Fig. 2 and Fig. 3, respectively. Over the southern pole, the NASADEM+ exhibits the expected highly variable deviations from the Copernicus GLO-90 (Fig. 3 (a)), while, in contrast, the Copernicus GLO-90 exhibits a relatively homogeneous negative shift of approximately 5 to 10 m relative to REMA (Fig. 3 (b)). Use of the ASTER GDEMv3 in the NASADEM+ provided a significant quality improvement over the DTED/GTOPO30 mix used in the OCODEM, but had its own quality issues as documented by Karlson et al. (2021) and Tachikawa et al. (2011). Figures 1, 2 and 3 demonstrate the increased age and resolution disagreements among the OCODEM, NASADEM+, and Copernicus GLO-90 over latitudes north of 60° N and Antarctica, as well as illustrating the relatively close agreement between the Copernicus GLO-90 and the ArcticDEM or REMA.

Figure 1 also shows differences between the NASADEM+ and the Copernicus GLO-90 over tropical and subtropical deciduous forests (i.e., the Amazon, central Africa, the southeast United States, and Indonesia). The NASADEM+ uses SRTMv4 data for the mid-latitude regions and the Copernicus GLO-90 uses data from the TanDEM-X satellite configuration, both of which are based on InSAR measurements, suggesting that the discrepancies over deciduous forests are probably not attributable to differences in radar wavelengths. Most likely, mid-latitude disagreement between NASADEM+ and Copernicus GLO-90 is the result of some combination of differences in the time of data acquisition, in data processing methods used in generating the DEMs, or in instrument deployment and data acquisition techniques (i.e, using radar instrumentation on a space-shuttle with attached receive-only antenna for SRTM as opposed to a constellation of two satellites for TanDEM-X) (Fahrland et al., 2020; Crippen et al., 2016; Simard et al., 2016; Farr et al., 2007). For the purposes of optimizing OCO-2 retrievals, the differences between the NASADEM+ and Copernicus GLO-90 over mid-latitude regions are generally too small to significantly change $X_{CO_2}$. Though the most significant differences relative to the OCODEM are over the polar regions, both the NASADEM+ and Copernicus GLO-90 include significant adjustments over mid and low latitudes relative to the OCODEM. Changes in the NASADEM+ relative to OCODEM at latitudes between $\pm 60°$ are a reflection of many advancements made in data processing, void filling, land/water masking, and other DEM generation techniques in the years since the first version of SRTM was released. Differences between the Copernicus GLO-90 and OCODEM reflect advancements in both DEM generation and data acquisition techniques via the TanDEM-X mission, which also uses more recent elevation measurements.

## 4.2 Spatial inconsistencies

There appears to be a small discontinuity across the 60° N parallel in the NASADEM+ with an average drop in elevation of approximately 4 m, which is not exhibited in the Copernicus GLO-90 (see Fig. 4). This discontinuity most likely results from the transition between the SRTMv4 and ASTER GDEMv3 data collections within the NASADEM+. This finding is consistent with results from Karlson et al. (2021), who report that ASTER GDEMv3 tends to underestimate elevations in Sweden relative to other DEMs and validation measurements, alluding to a possible negative bias over the northern high latitudes. If, as the results in Fig. 4 suggest, Copernicus has good continuity across the 60° N parallel, then the break at 60° N in panel (b) of Fig. 1 and panel (b) of Fig. 2 may indicate a discontinuity in the OCODEM across this boundary as well. There are also some cases of dry lake beds and other geologic features that have elevations below mean sea level (i.e., below the surface of the reference

ellipsoid, EGM96) which the Copernicus GLO-90 maps accurately, but are mapped as zero elevation in the NASADEM+. An investigation revealed that most of these errors were due to poor void filling or other algorithm errors in the OCODEM preprocessing of the SRTMv1 data that propagated through to the NASADEM+ database.

## 4.3   Impacts of altitude differences on $X_{CO_2}$

After updating the referenced DEM during the development of OCO-2 ACOS v11, it became apparent that changes in the DEM were dictating changes in dP, $dP_{frac}$, and $X_{CO_2}$, with particularly significant changes over northern high latitude regions (see Fig. 5 and 6). This is despite the fact that this update includes changes to other elements of the retrieval algorithm aside from a simple change to the DEM. We also see that the change in un-bias-corrected retrievals of $X_{CO_2}$ in Fig. 5 do not exhibit the same strong dependence on the difference in altitude. A positive shift in altitude yields a negative shift in a priori surface pressure, a positive shift in dP and $dP_{frac}$, and a positive shift in bias-corrected $X_{CO_2}$. Near sea level, for example, a +10 m shift in surface elevation yields a $\sim$ +0.4 ppm shift in $X_{CO_2}$. For the most part, the NASADEM+ has lower altitudes than the OCODEM across the northern high latitude regions (see Fig. 1 (a) and 2 (a)), with a mean difference of -9 m. These elevation changes correspond to shifts in the v11 bias-corrected $X_{CO_2}$ and dP relative to v10, shown in Fig. 6 (a) and (b). The Copernicus GLO-90 has more varied changes in altitudes across the northern high latitudes and other regions relative to the OCODEM (see Fig. 1 (b) and 2 (b)), hence shifts in $X_{CO_2}$ and dP from v10 to v11.1 do not exhibit the striking, largely homogeneous shifts isolated over the northern high latitudes that were observed when comparing v11 and v10 (see Fig. 6 (c) and (d)).

The area around the Bełchatów powerplant in Poland represents a remarkable example of how a more current and accurate DEM can reduce erroneous behaviour in $X_{CO_2}$. Near the powerplant, there is a large open-pit lignite mine that caused a false dipole in bias-corrected $X_{CO_2}$ observed in every target-mode OCO-2 overpass at the site for ACOS v11 and previous versions. Figure 7 shows that this dipole, present in OCO-2 v10 and v11 soundings, is almost entirely removed from the OCO-2 v11.1 soundings. This is most likely due to the fact that the TanDEM-X elevation data used in the Copernicus GLO-90 were collected much more recently than the SRTM data used in the NASADEM+ and OCODEM. Surface elevations are constantly changing due to natural and anthropogenic forces, and these changes can create significant errors in retrieved $X_{CO_2}$.

## 4.4   Changes in OCO-2 data throughput as a result of the v11.1 update

The update from v10 to v11 changes the percentage of retrievals that pass quality control filters. This can be attributed to a number of factors, including improvements in spectroscopy and radiative transfer modelling, as well as a better understanding of filtering parameters that allowed thresholds to be broader and more permissive. Differences in data throughput among OCO-2 versions, as a result of quality control thresholds, are mapped in Fig. 8. In these maps, v11.1 refers to OCO-2 v11.1 with the new and finalized quality control thresholds that account for bias corrected co2_ratio and h20_ratio parameters (see details in Appendix B), while v11.1[oQC] refers to the OCO-2 v11.1 retrievals with the original v11 quality control thresholds applied.

In the case of v11.1[oQC] changes in data throughput result only from changes to the values of altitude_stddev, dp_o2a (aka dP or $dP_{O_2A}$), dpfrac (aka $dP_{frac}$), and dp_abp (see parameter definitions in Table 2) as a direct consequence of changing the DEM without any adjustments to the quality control thresholds. Figure 8 (d) shows that in many regions changes in v11 had the desired effect of either maintaining a high fraction of data throughput or increasing the data throughput relative to v10, but data throughput decreased in tropical regions. This decrease in passable v11 data over the Amazon, central Africa, and Indonesia relative to v10 is mitigated in v11.1, as seen when comparing Fig. 8 (c) and (d). Figure 8 (e) and (f) show that the corrections to co2_ratio and h2o_ratio (Appendix B), and not the change in DEM, are primarily responsible for this recovery of lost tropical data in v11.1. The corrections to co2_ratio and h2o_ratio also yield some increases in data throughput over northern high latitudes. Figure 8 (e) demonstrates that the v11.1 change to the DEM and corresponding parameters listed in Table 2 yields some significant changes in the fraction of data throughput, mostly over the northern high latitudes, as well as mid-latitude regions with variable topography. Overall, v11.1 has much higher data throughput over northern high and mid latitudes relative to either v10 or v11, while sustaining minimal declines in data throughput over tropical regions. Tables 2 and 3 list the new and changed parameters in v11.1 that largely account for the observed changes in throughput.

## 4.5 Reduced $X_{CO_2}$ bias relative to TCCON

Retrievals from all three OCO-2 ACOS versions (v10, v11, and v11.1) were compared against TCCON measurements using methods described in Sect. 3.3, and all soundings were matched between versions to exclude the effects of changes in quality control filtering. Results in Table 5 show that the overall mean biases and standard deviations in bias progressively improve from v10 to v11.1, with the smallest mean biases and standard deviations in bias relative to TCCON observed in v11.1. As shown in Fig. 9, biases in OCO-2 retrievals relative to TCCON measurements have undergone small changes from v10 to v11 that mostly yield less intra-site variability in biases in v11. The update to v11.1 from v11 yielded relatively small changes in OCO-2 biases relative to TCCON at most sites, with the notable exceptions being the high northern latitude sites at Sodankylä and Eureka. While v11.1 does not uniformly reduce OCO-2 biases at all sites, overall, v11.1 yields reduced biases at most of the TCCON sites. Figures 10 and 11 show that there are generally higher elevations over the Sodankylä TCCON site in the OCODEM (used in v10), then a large negative shift in the NASADEM+ (used in v11), and finally, elevations somewhere in between the OCODEM and NASADEM+ in the Copernicus GLO-90 (used in v11.1). These elevation differences among DEMs over Sodankylä directly correspond to the large negative shift in OCO-2 biases from v10 to v11 and the positive adjustment back toward zero mean bias in v11.1. Figure 10 shows that $dP_{frac}$ also varies less over the Sodankylä target in v11.1, implying that the distribution of a priori surface pressures across the target, driven by variations in the underlying DEM, better match the retrieved surface pressures over the target. This tends to reduce scatter in $X_{CO_2}$. Additional target examples for Pasadena, Lauder, and Eureka can be found in the Sect. S3 in the supplemental materials, although these targets generally show smaller changes in altitudes across versions or, specifically in the case of Eureka, limited spatial coverage for target mode measurements after applying quality filtering. At Eureka we also see a similar pattern of large negative shift in bias from v10 to v11, with a correction back toward v10 biases in v11.1. At Eureka the shift from v11 to v11.1 actually moves the bias further from

**Table 5.** Overall bias and standard deviation in bias relative to TCCON measurements when combining data from all sites for target mode observations or combined land nadir and land glint (LNLG). Coincidence criteria and data handling are described in Sect. 3.3

| Sounding type | mean bias / ppm | standard deviation in bias / ppm |
|---|---|---|
| Target v10 | 0.31 | 0.80 |
| Target v11 | 0.15 | 0.74 |
| Target v11.1 | -0.04 | 0.73 |
| LNLG v10 | 0.07 | 0.98 |
| LNLG v11 | -0.06 | 0.98 |
| LNLG v11.1 | 0.01 | 0.97 |

zero, but is still less bias than is seen in v10. Due to the sparse data coverage at Eureka it is difficult to explore these changes in more depth, as we have been able to do with Sodankylä. Reduced biases in target mode soundings in OCO-2 v11.1 at Lauder and Pasadena are also remarkable (see Fig. 9 (b)) and may be at least partly explained by the improved representation of the highly variable topography around these sites with the Copernicus GLO-90. In general, we observe larger differences in mean OCO-2 biases among versions when comparing to OCO-2 target mode observations, which may suggest that errors related to the proximity of OCO-2 measurements to the site overwhelm changes from the DEM.

## 4.6 Impact of DEM-driven $X_{CO_2}$ differences on flux estimates

The impact of DEM-driven $X_{CO_2}$ differences on flux estimates was examined through a series of atmospheric $CO_2$ inversion analyses, using methods explained in Sect. 3.4. These experiments isolated the DEM-driven impacts by employing the v10 bias correction using a $dP_{frac}$ value adjusted for each DEM, but excluding other retrieval changes from the v11 update. Using NASADEM+ instead of the OCODEM resulted in a negative shift of -0.4 ppm in annual mean $X_{CO_2}$ over latitudes north of 60° N with more muted differences south of 60° N (see Fig. 12 (a) and (d)). This shift in $X_{CO_2}$ resulted in a meridional shift in the estimated $CO_2$ fluxes. The net sink north of 60° N was increased by 68 to 102 TgC (25 to 35 % of the sink) and a compensating reduction in uptake of 101 to 159 TgC (6 to 10 % of the sink) occurred over 30° to 60° N (see Fig. 12 (e) and (f)). This meridional shift is significant compared to uncertainties in carbon uptake over these latitude bans. For example, the v10 OCO-2 Modelling Intercomparison Project (MIP; Byrne et al., 2023) reports a standard deviation among inversions systems of 354 TgC for 30° to 60° N and 113 TgC for 60° to 90° N (LNLG experiment). In addition, this shift is significant relative to differences between experiments in the v10 OCO-2 MIP, as the median LNLG minus IS (in situ only) fluxes are -158 TgC for 30° to 60° N and 165 TgC for 60° to 90° N.

Differences are more muted when comparing estimated fluxes from retrievals using the Copernicus GLO-90 and OCODEM (see Fig. 12 (h)). Unsurprisingly, shifts in flux estimates exhibit similar spatial patterns to the shifts in $X_{CO_2}$ (see Fig. 12 (g) and (h)). The zonal mean shifts in fluxes using the Copernicus GLO-90 relative to fluxes using the OCODEM are not systematic when considering the spread among the three flux inversions (shown as the grey shading in Fig. 12 (c), (f), and (i)).

We know that similar changes to fluxes may be present on urban spatial scales ($< 100$ km) if the OCO-2 (or OCO-3) sounding elevations are inaccurately characterized due to DEM errors. However, the coarse resolution global model used here is not suitable for characterizing the impact on local or urban scale flux estimates and associated changes due to changing the DEM.

## 5 Conclusions

Accurate knowledge of surface pressures is essential for accurate calculations of $X_{CO_2}$ from atmospheric remote sensing observations. In the case of OCO-2, the a priori surface pressure is generally more accurate than the retrieved surface pressure, but cannot be constrained within the retrieval. Applying an empirical bias correction in post-processing as a function of the retrieved surface pressure error significantly improves $X_{CO_2}$, but necessitates the inclusion of a robust, consistent, and accurate DEM to inform the a priori surface pressure.

In this analysis, we have demonstrated significant shortcomings in the OCODEM, used for OCO-2 ACOS v10 retrievals and previous versions of the algorithm, as well as the NASADEM+, used in the first iteration of OCO-2 ACOS v11. Problems especially persist in the NASADEM+ over high northern latitudes (ASTER GDEMv3) and Antarctica (RAMPv2). Through this analysis and other evaluations of the Copernicus GLO-90 (Li et al., 2022; Marešová et al., 2022; Carrera-Hernández, 2021; Guth and Geoffroy, 2021; Karlson et al., 2021; Fahrland et al., 2020) there is substantial evidence to suggest that the Copernicus GLO-90 is the most globally continuous and accurate global DEM that fits the requirement of the OCO-2 ACOS retrieval algorithm for a globally consistent, void-free, and publicly available surface terrain model.

We show that changes to the DEM in the OCO-2 ACOS retrieval algorithm have significant impacts on the bias-corrected $X_{CO_2}$ product, which is a direct result of the inclusion of the $dP_{frac}$ term in the OCO-2 bias correction (see Fig. 5). DEM changes on the order of 10 m correspond to shifts in $X_{CO_2}$ on the order of 0.4 ppm (see Fig. 1 and 6). Relative to the OCODEM, changes are largest for the NASADEM+ north of $60°$ N, with smaller differences elsewhere and for the Copernicus GLO-90. Differences in $X_{CO_2}$ introduced by the NASADEM+ have a significant impact on flux inversion analyses, driving meridional shifts in carbon uptake of $\sim 100$ TgC over $30°$ to $60°$ N and $60°$ to $90°$ N. This is comparable in magnitude to differences between the IS and LNLG experiments in the v10 OCO-2 MIP Byrne et al. (2023).

As a result of this study, the OCO-2 team has produced OCO-2 v11.1, which is an update to OCO-2 lite files including the implementation of the Copernicus GLO-90 globally, and corresponding changes to other retrieval parameters, as described in Sect. 2.1 and 3.2. The v11.1 product is found to be more accurate than v11, and should supersede v11 in all instances. Overall, OCO-2 v11.1 retrievals provide increased data throughput after quality control filtering relative to v11 and v10. OCO-2 v11.1 retrievals have improved accuracy relative to TCCON at Sodankylä when compared to v11 and v10 retrievals, while excluding the effects of differences in quality control filtering between versions (see Fig. 10). Overall, mean biases and standard deviations in bias relative to TCCON are the smaller in v11.1 than in either v10 or v11 (see Table 5).

We encourage other satellite missions that use a DEM as supplemental data in their trace gas retrievals to consider the possible impacts their choice of DEM may have on their data product. Retrievals of trace gases with long atmospheric lifetimes such as $X_{CO_2}$ and $X_{CH_4}$ are the most sensitive to inaccuracies in surface pressure that could be driven by the underlying DEM. We found that DEM errors in the 10 to 20 m range can yield errors in the dry air column of 0.1 to 0.2 %. For trace gases with much larger variability (e.g., CO), these discrepancies would be less significant.

*Code and data availability.* OCO-2 Lite files are produced by the NASA OCO-2 project at the Jet Propulsion Laboratory, California Institute of Technology, and are available from the NASA Goddard Earth Science Data and Information Services Center (GES-DISC; https://daac. gsfc.nasa.gov/). TCCON data are available from the TCCON data archive, hosted by CaltechDATA: https://tccondata.org/. References to data from individual TCCON sites are also listed in Table 4. The NASADEM+ was accessed through a private archive operated by the Jet Propulsion Laboratory, California Institute of Technology. The Copernicus GLO-90 was accessed through Amazon Web Services, and is also available from Copernicus by visiting https://spacedata.copernicus.eu/collections/copernicus-digital-elevation-model and creating an account. The ArcticDEM and REMA are made publicly available by the Polar Geospatial Center, University of Minnesota at https: //www.pgc.umn.edu/data/.

## Appendix A:  Additional updates in OCO-2 ACOS v11

Compared to the changes between ACOS v8/9 and v10, the updates to the ACOS v11 level 2 full physics retrieval (L2FP) algorithm had a modest effect on the $X_{CO_2}$ estimates. One of the primary changes is the use of the ABSCO v5.2 tables, which included updates to the $CO_2$ line mixing in the strong $CO_2$ band and updates to the $H_2O$ line parameters in both the strong $CO_2$ and weak $CO_2$ bands (Payne and Oyafuso, 2022; Payne et al., 2022).

Another important update to v11 is the NOAA data used for the $CO_2$ prior. Starting in v10, the priors use NOAA data from Mauna Loa and American Samoa to set the secular growth rate. In v10, the NOAA data used ended in 2018 and was extrapolated to time periods beyond 2018. In v11, the priors use NOAA data obtained more frequently, with a latency of $\sim 1$ month (Sect. 4, Laughner et al. 2023). Without this change, the prior value would diverge from reality over time due to the

ever-increasing secular trend and rapid deviations from the secular trend due to extreme events, such as ENSO events, wildfires, or volcanic activity (Liu et al., 2017; Chatterjee et al., 2017; Gurney et al., 2012).

575

A number of very minor updates to the L2FP radiative transfer modules were made for v11, none of which appeared to have significant effect on the $X_{CO_2}$ estimates. First, the land surface Bidirectional Reflectance Distribution Function (BRDF) model was updated. The land surface BRDF was originally implemented in v8 (see Sect. 3.4 and Appendix B of O'Dell et al., 2018). During the development of v8, a multiple component kernel was implemented, which included an unpolarized BRDF kernel (the RPV kernel; Rahman et al., 1993), and a polarized BRDF kernel based on the soil surface model described in Maignan et al. (2009). Testing during v8 development showed that the polarized kernel was not needed, and the operational v8 was intended to use the unpolarized RPV kernel for the land surface BRDF. Due to a coding error, the polarized kernel was still present with a small amplitude relative to the RPV kernel within the ACOS land surface model in v8 and v10. In v11, this error was corrected and the land surface BRDF is now the unpolarized RPV kernel as described in O'Dell et al. (2018). Overall, the changes in $X_{CO_2}$ from this update were very marginal.

A minor update to v11 affects retrievals over water surfaces, where the surface model was changed from a Cox & Munk parameterization with a per-spectral band additive Lambertian component (Nelson et al., 2020), to a per-spectral band scaled Cox & Munk parameterization. Although this change had little effect on the $X_{CO_2}$ estimates, it significantly improved the accuracy of the retrieved wind speed, and produced an overall more linear behaviour with reduced dependence on the prior. Another small fix was to correct an error in a sign convention in the Stokes-U component of the Cox & Munk water surface model. The wind speed retrievals are now consistent across all 8 footprints, whereas in earlier versions there was a linear bias across the footprints. Finally, the LInearized Discrete Ordinate Radiative Transfer (LIDORT) modules were updated to the newest version (v3.8 for LIDORT) (Spurr and Christi, 2019).

595

The source of the prior meteorology has not been updated since ACOS v8/9. It still relies on the GMAO Goddard Earth Observing System (GEOS) Forward Processing Instrument Team (FPIT) product in both v10 and v11. However, the GMAO is currently in the process of phasing out the FPIT product and replacing it with a new product (GEOS-FPIT). A switch over to the new IT meteorology product will occur in 2024.

600

A number of changes were made to level 1B (L1B) calibration in v11 as well. Use of a new stray light correction applied to pre-flight data yielded updated pre-flight gain as well as instrument lineshapes (ILSs). The background noise model was updated based on in-flight data, whereas previously only pre-flight data was used. Further changes included updated relative radiometry based on improved lunar analysis, adjusted gain degradation, improved dispersion trending, and identification of additional bad spectral samples. These improvements resulted in mostly minor impacts to the final, bias-corrected $X_{CO_2}$.

## Appendix B: Improved Filtering Variables: co2_ratio and h2o_ratio

As described in Taylor et al. (2016, hereafter T16), two important filtering variables represent the ratios of retrieved gas columns from a simple, non-scattering retrieval called the IMAP-DOAS Preprocessor (IDP) from the strong $CO_2$ band relative to that from the weak $CO_2$ band. This ratio is calculated for two gases, $CO_2$ and $H_2O$, and the resulting variables are termed co2_ratio and h2o_ratio, respectively. These are somewhat analogous to the dp_abp variable from the A-Band Preprocessor, also described in T16. The latter represents the retrieved surface pressure relative to the prior surface pressure, as retrieved from a clear-sky, single-band $O_2A$ band retrieval. As described in T16, dp_abp was found to have a dependence on the solar zenith angle (SZA), and performs better after a piecewise linear bias correction relative to SZA is applied. No such bias correction was ever performed for co2_ratio or h2o_ratio.

Recent analysis has shown that both IDP gas column ratios do in fact suffer from minor biases. This is exemplified in Fig. B1, which shows the effect of biases in co2_ratio (panel a) and h2o_ratio (panel b). Fig. B1(a) shows a cloud-free OCO-2 overpass of a dark tropical forest in Liberia. Even though Aqua-MODIS shows the scene to be very clear, most soundings were marked as bad in v11, with most soundings failing the co2_ratio test. It was found that clear scenes over dark surfaces (such as tropical forests in Amazonia, Africa, and Southeast Asia) often failed this test, which was traced to a bias in co2_ratio as a function of the retrieved surface reflectivity in band 3 to band 2 (hereafter Aratio32). We similarly discovered a low bias in the retrieved h2o_ratio over very dry regions, such as in high latitudes; an example of this is shown in Fig. B1(b).

Plots of these "clear-sky" biases are shown in Fig. B2, for the co2_ratio as a function of Aratio32, and for the h2o_ratio as a function of TCWV. Additionally, as shown in Fig. B2(b), a unique problem with the h2o_ratio is that the quantity itself becomes more and more uncertain when the h2o column (i.e., TCWV) becomes very low ($\lesssim 10$ kg m$^{-2}$). It was found that the h2o_ratio bias in particular was much stronger in OCO-2 v10 IDP fits. Because the only significantly difference in the IDP algorithm between these versions was due to changes in spectroscopy, it became clear that these biases are due to spectroscopy errors. As described in Appendix A above, the water vapour spectroscopy was particularly improved in version 11 (ABSCO 5.2) relative to version 10 (ABSCO 5.1).

In order to correct for these biases, we characterized them as piecewise linear functions of Aratio32 for the co2_ratio and of $\ln(\text{TCWV})$ for the h2o_ratio. We fit the noise-driven uncertainty ($1\sigma$) in h2o_ratio similarly. The fit points are given in Table B1. We then constructed bias-corrected versions of co2_ratio and h2o_ratio as follows:

$$\text{co2\_ratio\_bc} = 1 + (\text{co2\_ratio} - \text{co2\_ratio\_bias}) \tag{B1}$$

$$\text{h2o\_ratio\_bc} = 1 + (\text{h2o\_ratio} - \text{h2o\_ratio\_bias}) \left( \frac{\text{h2o\_ratio\_uncert\_high\_TCWV}}{\text{h2o\_ratio\_uncert}} \right) \tag{B2}$$

**Table B1.** Points for co2_ratio and h2o_ratio piecewise linear fits

| co2_ratio piecewise linear fit | | | | | |
|---|---|---|---|---|---|
| x: Aratio32 | 0.05 | 0.2 | 0.4 | 0.8 | 1.2 |
| y: co2_ratio_bias | 1.0342 | 1.0275 | 1.0185 | 1.0110 | 1.0090 |

| h2o_ratio piecewise linear fit | | | | | | |
|---|---|---|---|---|---|---|
| x: TCWV[1] [kg m$^{-2}$] | 0.5 | 2 | 5 | 10 | 25 | 75 |
| y: h2o_ratio_bias | 0.94 | 0.9583 | 0.9964 | 1.017 | 1.025 | 1.025 |
| y: h2o_ratio_uncert | 0.08 | 0.073 | 0.0372 | 0.023 | 0.017 | 0.016 |

[1] The piecewise linear fit is done in ln(TCWV).

where co2_ratio_bias and h2o_ratio_bias represent the piecewise linear fits shown in Fig. B2, and similarly h2o_ratio_uncert is the piecewise linear fit to the $1\sigma$ standard deviation in the observed h2o_ratio. h2o_ratio_uncert_high_TCWV represents the asymptotic value of the uncertainty in h2o_ratio at high TCWV and is set to 0.016. The inclusion of this uncertainty ratio in the bias corrected $H_2O$ ratio is performed in order to give the h2o_ratio less weight in dry scenes where the $H_2O$ ratio is highly uncertain. This term is not necessary for co2_ratio, as its uncertainty does not significantly. Using these bias-corrected gas column ratios allows simple fixed thresholds to be used for quality test; the resulting filtering is significantly more robust, especially in cases of low Aratio32 or low TCWV. The improvements are shown in panel (f) of Fig. 8, which demonstrates that these changes led to increased throughput in the tropical forests (low Aratio32) as well as the high northern land areas (low TCWV).

*Author contributions.* Nicole Jacobs composed this manuscript and conducted the analysis under the supervision of Christopher O'Dell. Thomas Taylor helped to compose the section on OCO-2 v11 updates. Thomas Logan provided information on the DEMs, particularly the NASADEM+, and offered valuable insights into how DEMs are used in the broader context of NASA missions. Brendan Byrne perform the flux inversion analysis. Matthäus Kiel conducted the analysis of OCO-2 bias relative to TCCON. Rigel Kivi and Pauli Heikkinen are the principal investigators of the Sodankylä TCCON site and their observations were an essential component of our more in depth evaluation of OCO-2 biases over the northern high latitude regions. Aronne Merrelli aided in DEM data acquisition, played a key role in the OCO-2 ACOS v11 updates, and provided essential guidance on DEM differences. Vivienne Payne is the Lead Project Scientist for the OCO-2 mission and has overseen this analysis and all recent updates to the OCO-2 ACOS retrieval algorithm. Abhishek Chatterjee is the Deputy Project Scientist for the OCO-2 mission and principal investigator on the project that funded this work. He has supervised this work from the initial analysis to implementing the OCO-2 v11.1 update and the flux analysis conducted by Brendan Byrne. All coauthors have provided essential feedback, insights, and contributions to the content and verbiage of the manuscript in its entirety.

*Competing interests.* At least one of the (co-)authors is a member of the editorial board of Atmospheric Measurement Techniques.

*Disclaimer.* Reference herein to any specific commercial product, process, or service by trade name, trademark, manufacturer, or otherwise, does not constitute or imply its endorsement by the Authors or their Affiliation.

*Acknowledgements.* The authors acknowledge Rob Rosenberg for work with the noise model, ILS updates, and "c_background" change in the OCO-2 v11 update, as well as reviewing explanations in this manuscript. The authors acknowledge Josh Laughner for work with implementing GGG priors in OCO-2 v11 and clarifying the explanations in this manuscript. The authors acknowledge Fabiano Oyafuso, Brian Droiun, and Geoff Toon for work developing the ABSCO v5.2 spectroscopy tables used in OCO-2 v11. The authors acknowledge the PIs and others that operate the TCCON sites for their work collecting, processing, and publicly providing high resolution and high quality observations of atmospheric trace gas concentrations from all over the world. References to TCCON data used in this paper are listed in Table 4. This work is funded by NASA OCO Science Team grant # 80NSSC20K0006 as part of a larger collaborative project, "Diagnosing and attributing Arctic-Boreal carbon fluxes using in situ and satellite $CO_2$ monitoring network". Part of the research was carried out at the Jet Propulsion Laboratory, California Institute of Technology, under a contract with the National Aeronautics and Space Administration (80NM0018D0004).

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

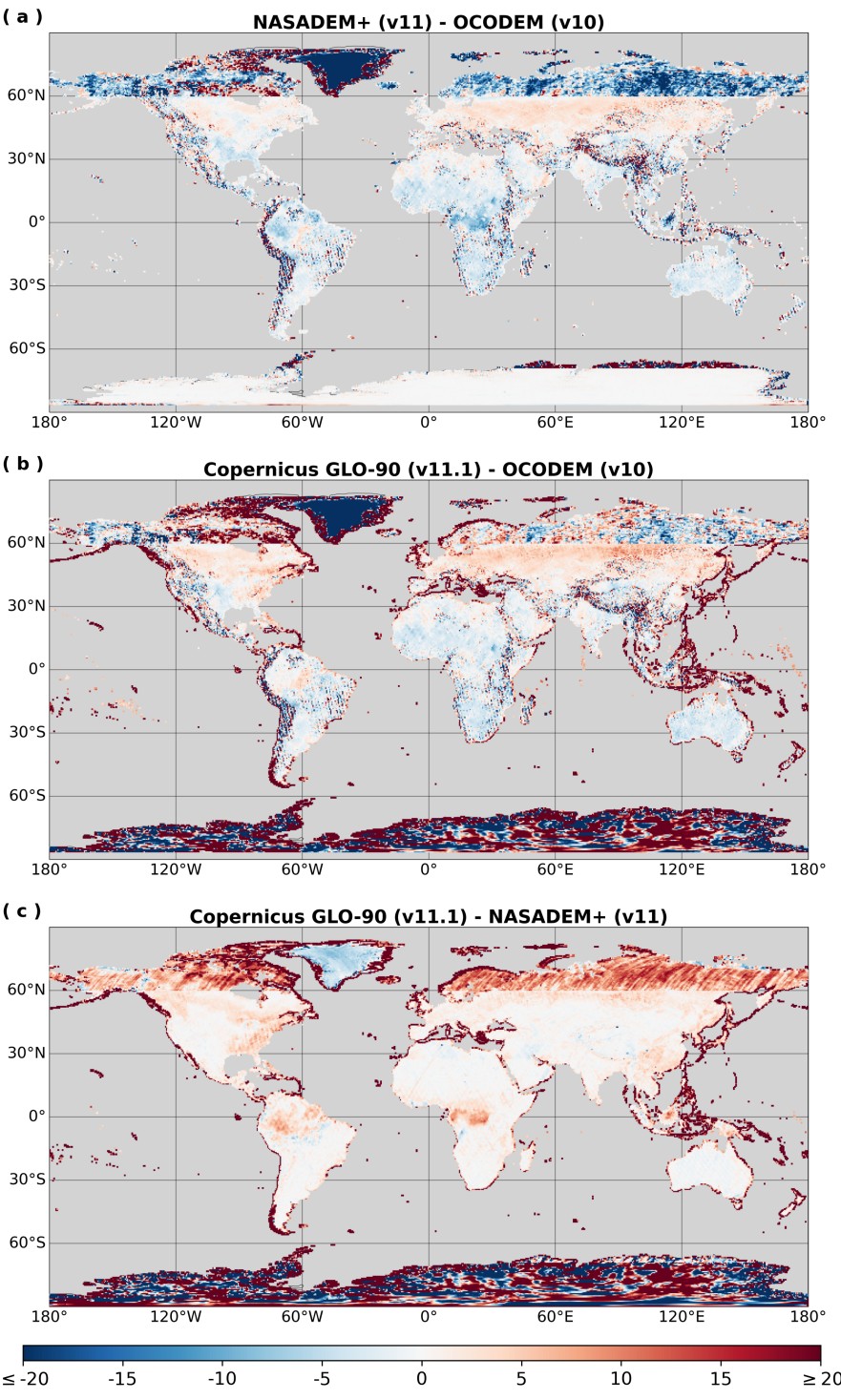

**Figure 1.** Global map of differences in elevation among OCODEM, NASADEM+, and Copernicus GLO-90 at $0.5° \times 0.5°$ resolution. The NASADEM+ collection utilized the older RAMPv2 DEM data in Antarctica, and the ASTER GDEMv3 data in most areas north of $60°$N.

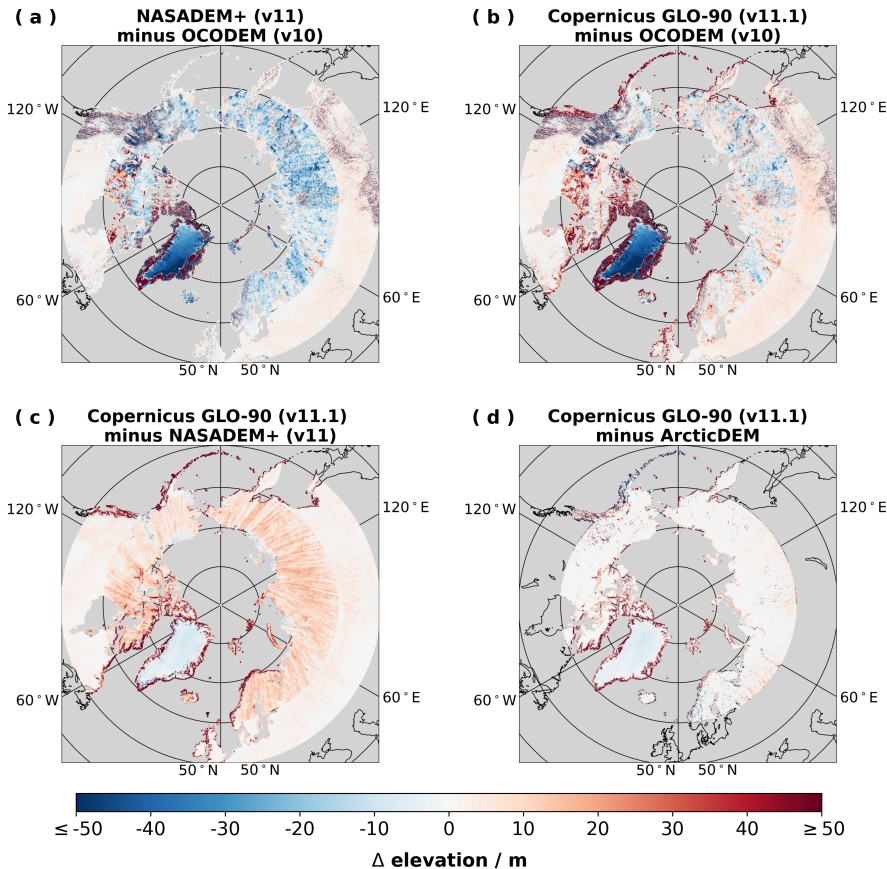

**Figure 2.** Maps of differences in elevation over northern high latitude region among the OCODEM, NASADEM+, ArcticDEM, and Copernicus GLO-90 at $0.1° \times 0.1°$ resolution. Refer to Sect. 2.3 to 2.6 for descriptions of the DEM data.

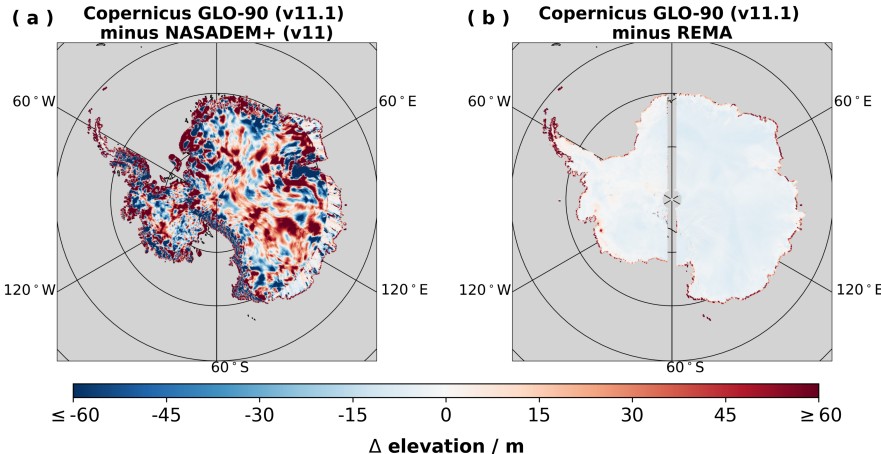

**Figure 3.** Maps of differences in elevation over Antarctica among the NASADEM+, Copernicus GLO-90, and REMA at $0.1° \times 0.1°$ resolution. The NASADEM+ collection utilized the older RAMPv2 DEM data in Antarctica.

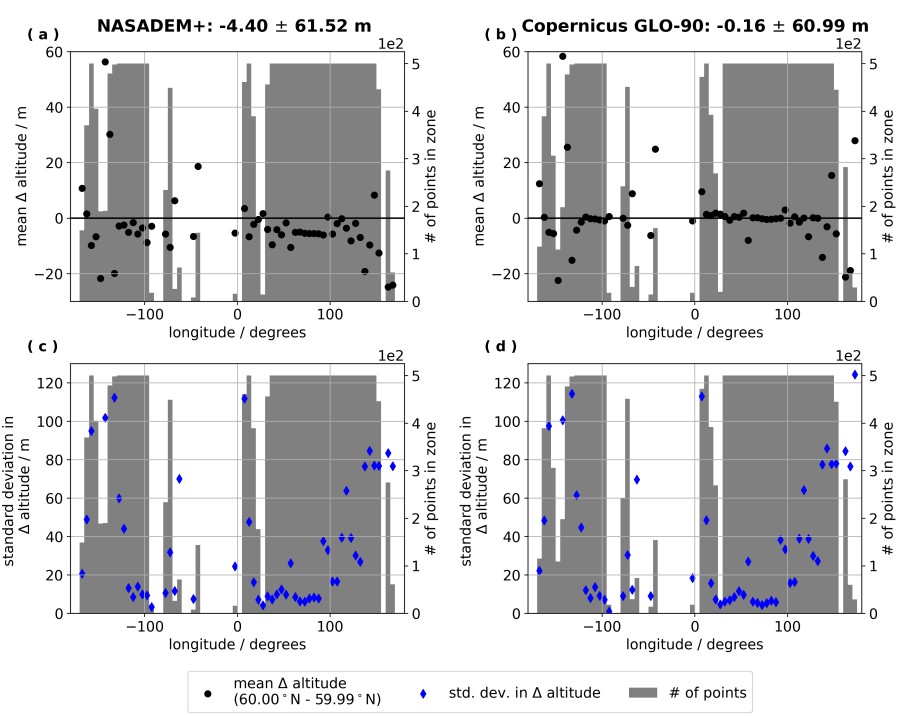

**Figure 4.** Shifts in DEM elevations crossing the 60° N parallel in the NASADEM+ and Copernicus GLO-90. For these plots the average elevations are taken from each DEM within each 0.01° latitude by 5° longitude region and the difference is taken as the region north (60.00° N to 60.01° N) of 60° N minus the region south (59.99° N to 60.00° N) of 60° N. Left, (a) and (c), are the NASADEM+ and right, (b) and (d) are the Copernicus GLO-90. Above each plot is the global average ± one standard deviation of all zonal differences across this parallel for each DEM. Also shown are the number of DEM data points in each 5° zonal bin (i.e., there are fewer DEM data points where there is ocean). Some differences are a real result of topographical changes (commonly corresponding to higher standard deviations in altitude, shown in (c) and (d)), but one would not expect to see a global average difference when crossing this parallel.

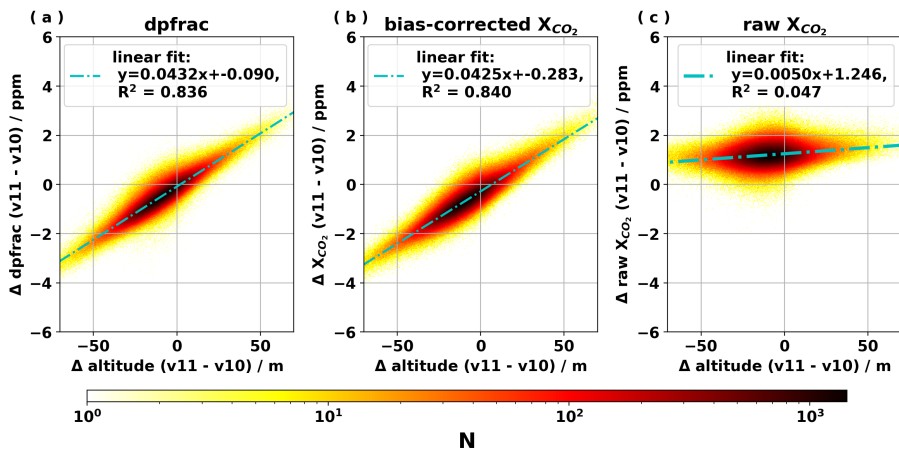

**Figure 5.** Correlations between change in (a) dP$_{frac}$, (b) bias-corrected X$_{CO_2}$, and (c) raw retrieved X$_{CO_2}$ with respect to change in altitude when updating from OCO-2 v10 to v11.

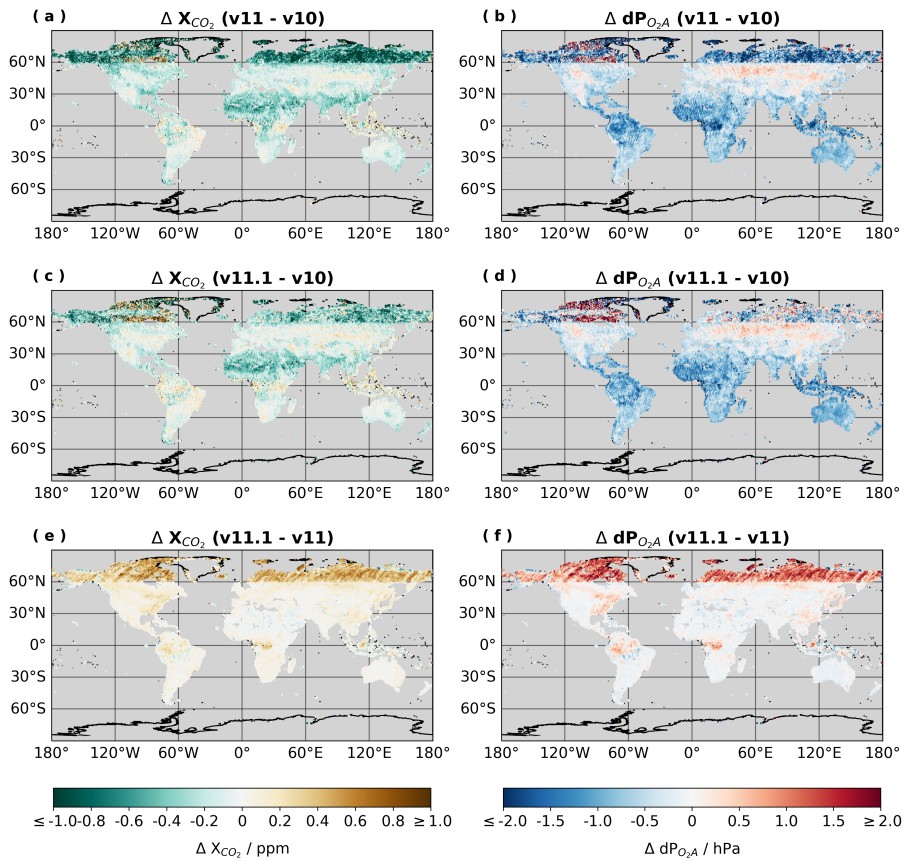

**Figure 6.** Global map of differences in the bias-corrected $X_{CO_2}$ product and $dP_{O_2A}$ (retrieved minus a priori surface pressure in the $O_2A$ band) among OCO-2 retrieval versions v10, v11, and v11.1 with $0.5° \times 0.5°$ resolution. These maps include observations made between September 2014 and February 2022.

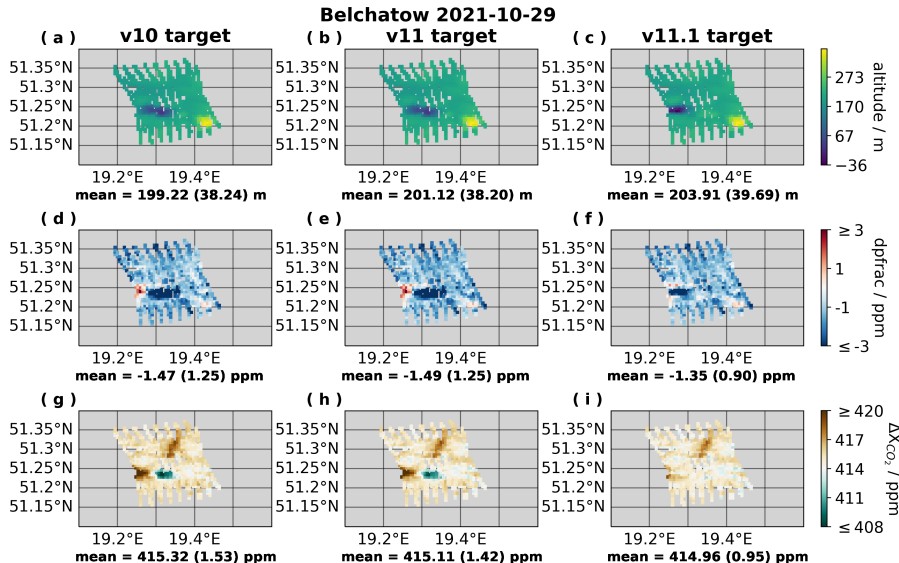

**Figure 7.** Sounding altitudes (top row; calculated for the sounding footprint as described in Sect. 3.1), retrieved $dP_{frac}$ (middle row), and OCO-2 bias-corrected $X_{CO_2}$ (bottom row) during an OCO-2 target-mode overpass of the Bełchatów powerplant on 29 October 2021. Plots (a), (d), and (g) show results with OCO-2 v10 retrievals. Plots (b), (e), and (h) show results with OCO-2 v11 retrievals. Plots (c), (f), and (i) show results with the OCO-2 v11.1 retrievals. No quality control filters are applied in this case, but soundings are matched between OCO-2 ACOS versions.

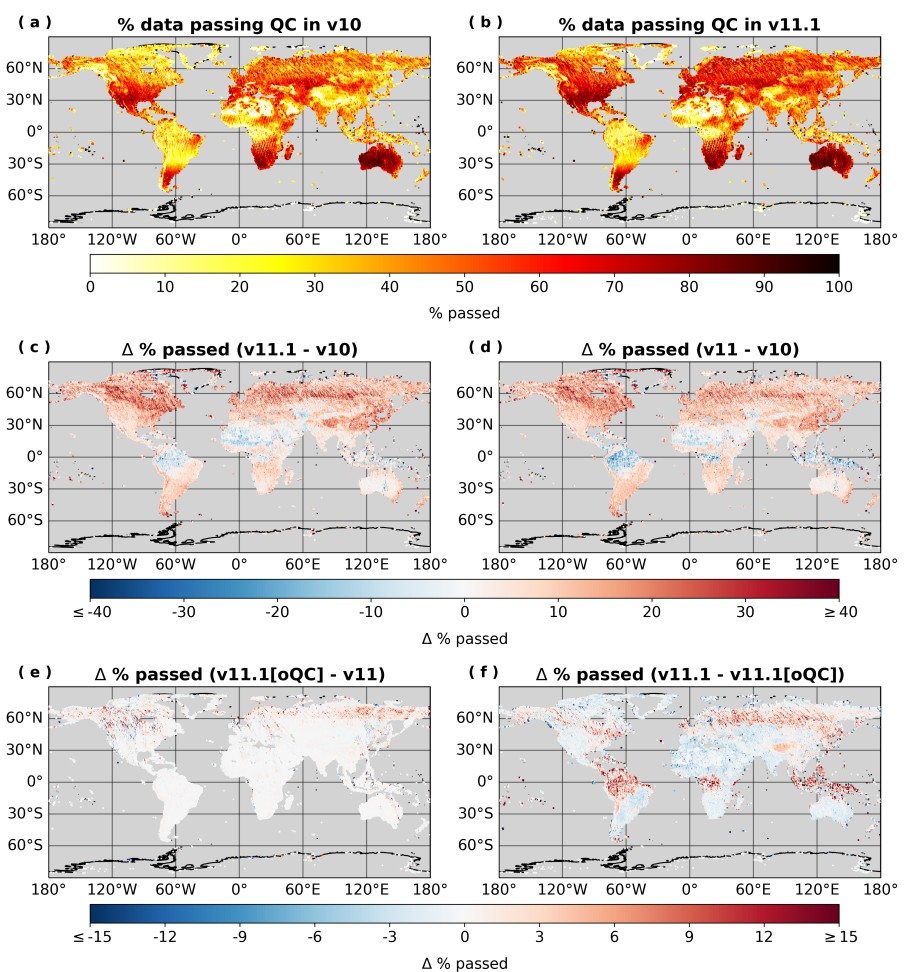

**Figure 8.** Maps of (a) fractional data throughput (% of lite file soundings that pass quality controls) in v10; (b) fractional data throughput in v11.1; (c) the change in fractional data throughput to v11.1 from v10 (% data passed in v11.1 minus % data passed in v10); (d) the change in fractional data throughput to v11 from v10; (e) the change in fractional data throughput with the update from v11 to v11.1 without changes to quality control thresholds (% data passed in v11.1 minus % data passed in v11); (f) the change in fractional data throughput due to other changes in quality control filtering in v11.1, most notably the corrections on h2o_ratio and co2_ratio (see details in Appendix B). In this figure, v11.1 refers to OCO-2 v11.1 with the new and finalized quality control thresholds that account for bias corrected co2_ratio and h20_ratio parameters, while v11.1[oQC] refers to the OCO-2 v11.1 retrievals with the original v11 quality control thresholds applied. These maps include observations made between June of 2015 and February of 2022, aggregated to $0.5° \times 0.5°$ resolution.

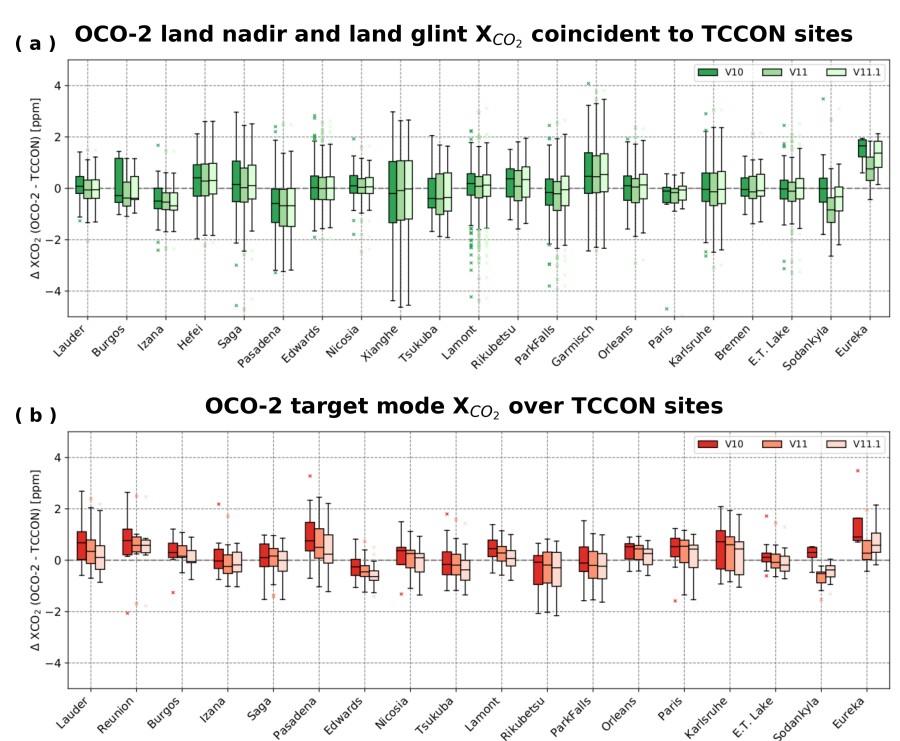

**Figure 9.** Boxplots of OCO-2 bias relative to TCCON measurements for OCO-2 v10, v11, and v11.1 (see Sect. 2.2 for coincidence criteria and Table 4 for site details). Plot (a) includes all land nadir and land glint OCO-2 soundings coincident to each ground site and plot (b) includes target mode OCO-2 soundings. For each site sounding matching is performed so that only soundings that meet quality control criteria for all three OCO-2 ACOS versions are included.

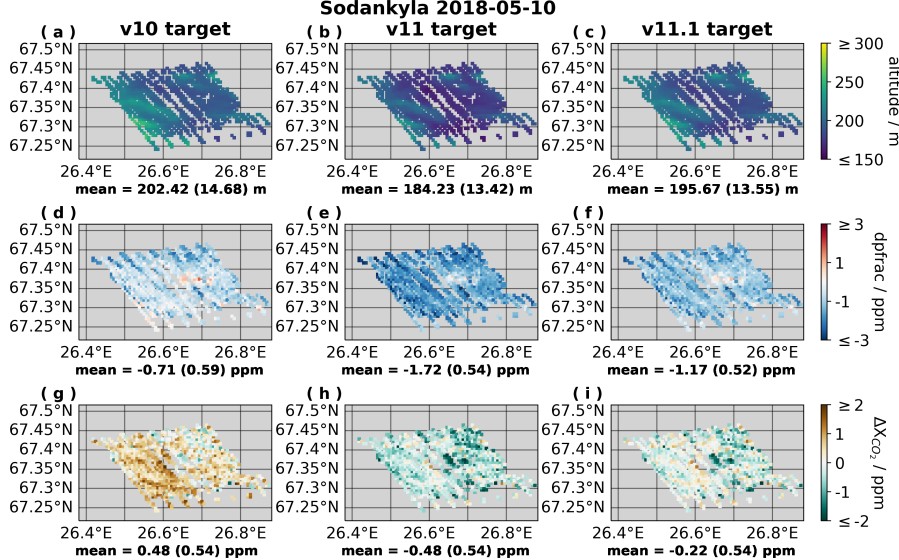

**Figure 10.** Sounding altitudes (top row; calculated for the sounding footprint as described in Sect. 3.1), retrieved $dP_{frac}$ (middle row), and OCO-2 bias in bias-corrected $X_{CO_2}$ relative to TCCON $X_{CO_2}$ (see Sect. 3.3) (bottom row) during an OCO-2 target-mode overpass at Sodankylä on 10 May 2018. Plots (a), (d), and (g) show results with OCO-2 v10 retrievals. Plots (b), (e), and (h) show results with OCO-2 v11 retrievals. Plots (c), (f), and (i) show results with the OCO-2 v11.1 retrievals. Only soundings that pass quality control filters in all three versions of the OCO-2 retrievals are included, such that the same set of soundings are included in all plots.

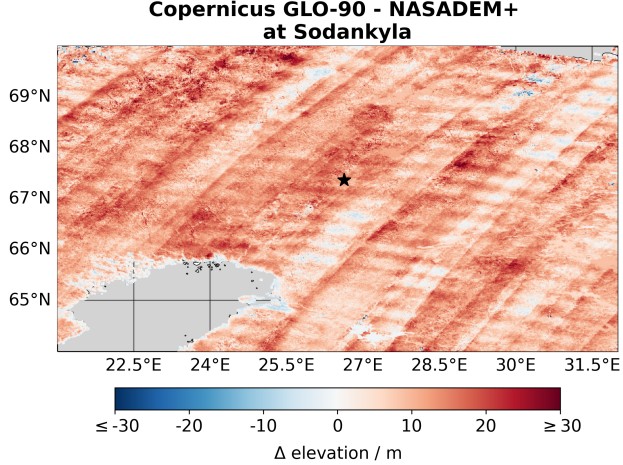

**Figure 11.** Map of the differences in elevation between the Copernicus GLO-90 and NASADEM+ in the area around the TCCON site at Sodankylä, Finland (site location shown at the black star).

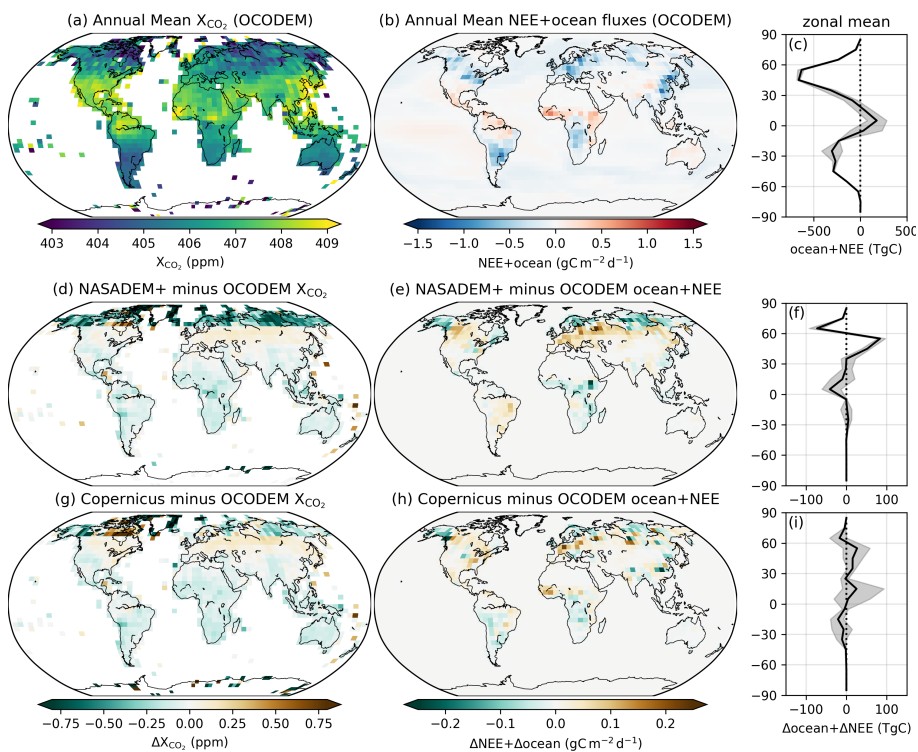

**Figure 12.** Surface CO$_2$ fluxes estimated from ACOS v10 data with the OCODEM, NASADEM+, and Copernicus GLO-90. The top row shows results from the original v10 retrievals: (a) map of annual mean X$_{CO_2}$ at $4° \times 5°$ resolution; (b) map of estimated annual mean NEE+ocean at $4° \times 5°$ resolution; (c) zonal mean of NEE+ocean fluxes with the spread due to different model priors shown as grey shading. The middle row shows the difference between results from v10 retrievals using the NASADEM+ (see Sect. 3.4) and the original v10 retrievals using the OCODEM: (d) map of $\Delta$X$_{CO_2}$ = v10[NASADEM+] - v10[OCODEM]; (e) map of $\Delta$ NEE+ocean = v10[NASADEM+] - v10[OCODEM]; (f) zonal mean of $\Delta$ NEE+ocean. The bottom row shows the difference between results from v10 retrievals using the Copernicus GLO-90 and the original v10 retrievals using the OCODEM: (g) map of $\Delta$X$_{CO_2}$ = v10[Copernicus] - v10[OCODEM]; (h) map of $\Delta$ NEE+ocean = v10[Copernicus] - v10[OCODEM]; (f) zonal mean of $\Delta$ NEE+ocean.

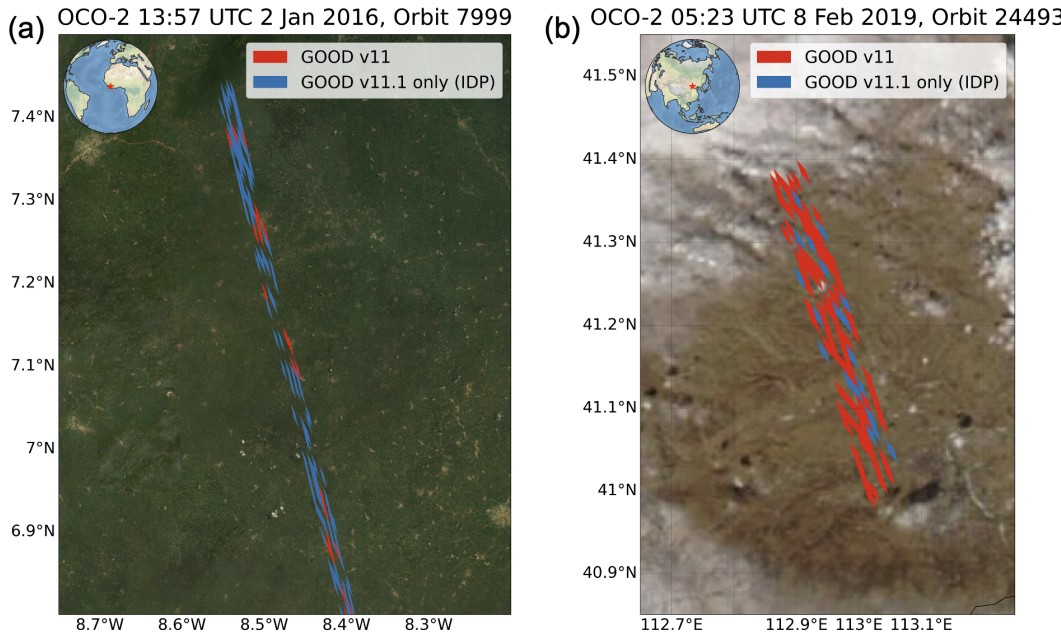

**Figure B1.** Quality flagging for two select OCO-2 overpasses. (a) Liberia on January 2, 2016 showing the effect of co2_ratio changes. Simultaneous Aqua-MODIS imagery (background of the figure) shows the case to be cloud-free. The standard v11 OCO-2 quality flag, using raw values of co2_ratio and h2o_ratio, passes as clear only a few soundings (red). Using the bias-corrected gas ratios, the updated v11.1 quality flag passes most soundings in this case (blue). The mean albedo in the strong CO2 band is about 0.05, and identifies this as a very dark scene, especially challenging the co2_ratio. (b) Same as (a), but shows a section of Inner Mongolia, China, on February 8, 2019, near the city of Ulanqab (bottom right of the image). There are some clouds in the northern and southern sections of the scene, but the inner part is relatively clear according to Aqua-MODIS. This scene is extremely dry, with TCWV $\sim$1.6 kg m$^{-2}$, and highlights the effect of the h2o_ratio filtering change. The improvement is much more modest than for co2_ratio, but does pass 10 to 20% more soundings in very dry scenes.

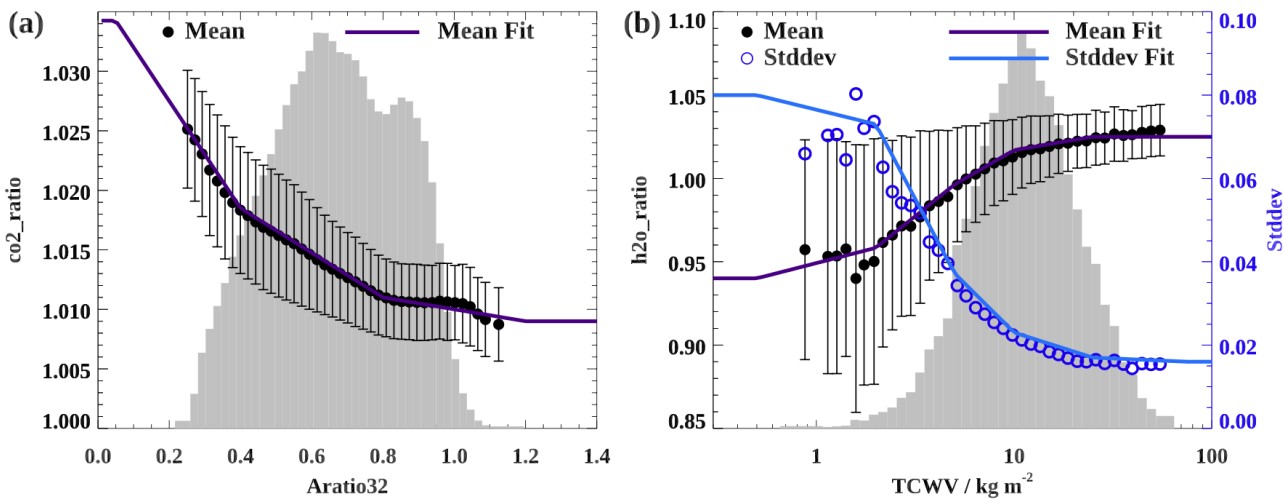

**Figure B2.** "Clear-sky" biases in co2_ratio and h2o_ratio and associated fits as functions of Aratio32 and TCWV, respectively. "Clear-sky" soundings were determined by requiring soundings to pass all xco2_quality_flag tests except for the co2 and h2o ratio tests, for a globally-representative set of OCO-2 land soundings spanning 2014 through 2022. Results for ocean soundings were nearly identical. (a) Mean co2_ratio for nearly-clear sky soundings as a function of Aratio32, defined as albedo_sco2/albedo_wco2. The $1\sigma$ standard deviation in each Aratio32 bin is given by the vertical error bars. The relative distribution of Aratio32 is shown in grey. The piecewise linear fit to the co2_ratio bias is shown as the thin purple line ("Mean Fit"). (b) Same as (a), but showing the h2o_ratio bias as a function of total column water vapour (TCWV). The open dark blue circles show the $1\sigma$ standard deviation in h2o_ratio, with the corresponding piecewise linear fit shown in light blue ("Stddev Fit").