# Peer review of "The importance of digital elevation model accuracy in $X_{CO_2}$ retrievals: improving the OCO-2 ACOS v11 product"

_Atmospheric Measurement Techniques, 2023_

## Referee Comment (RC1)

**1  General remarks**

The manuscript by Jacobs et al. describes updates to the OCO-2 ACOS v11 product with a focus on the improvements gained by using an updated digital elevation model. They compare various DEMs with a focus on the high latitudes and investigate the improvements gained by comparison to TCCON and the impact on flux estimates. The manuscript is overall well written, however some areas could benefit from additional information.

**Data description of OCO-2 retrievals**

In the manuscript measurements taken in target, land nadir and land glint mode are mentioned and used. Please provide some explanation on the properties and differences between measurements taken in these modes. This would be especially important in context of the results presented in Table 6. Why was the initial bias in v10 lower in LNLG measurements compared to target mode measurements?

**Inversion results**

In section 4.6 the impact of the DEM on flux estimates is discussed by comparing both the NASADEM+ and Copernicus DEM fluxes to the OCODEM fluxes. To me it seems that no significant difference in zonal mean fluxes are visible when using the Copernicus DEM. When using NASADEM+ the difference in zonal mean flux is however clearly visible. I think this is not sufficiently discussed in the paper and caused some confusion for me.

1. In the abstract and conclusion the stronger differences when using NASA-DEM+ are highlighted. In v11.1 the NASADEM+ is however replaced by the Copernicus DEM which shows no such differences. This seems confusing to me and this highlighting needs more justification (or it should be removed). At first glance this can be interpreted as an exciting result gained from using a new DEM (compared to v10). But I suppose this is meant to discourage the usage of v11 in comparison to v11.1?

2. In Figure 13 panels (e) and (h) look very similar, while (f) and (i) (or (d) and (g)) do not. Why is this the case?

3. In Figure 13 the maps are shown with a 4°x5° resolution, which is coarser than the resolution used when comparing DEMs. Why is this the case? Is this the resolution of the flux inversion? This should be clarified and also mentioned in section 3.4.

**2 Specific comments**

- Page 1 Line 10: Why highlight the flux differences when using NASA-DEM+, when you decide on using the Copernicus DEM in the updated version (which does not show significant flux differences)?

- Page 2 Line 24: Could you shortly specify what kind of flaws?

- Page 2 Line 26: Please specify what kind of effects the fixed surface pressure would have. Was the retrieval with a fixed surface pressure informed by an updated DEM tested?

- Page 3 Line 1: It would be helpful to provide a brief summary of the results from Hachmeister et al. (2022).

- Page 3 Line 22: Is this relevant for you comparisons between different DEMs for the XCO2 retrieval? I don't think you follow up on this later in the manuscript.

- Page 4 Line 31: Why these two gases in particular? Would other trace gases be similarly affected?

- Page 5 Table 1: I am not sure whether this table is necessary since it contains no additional information.

- Page 6 Line 24 – 25: To me it is not clear what motivates these formulas. Where does the 0.016 come from and what is meant by: "co2_ratio_bias and h2o_ratio_bias represent the piecewise linear fits"?

- Page 8 Line 1-2: How was the averaging performed? What is the order of magnitude of these artifacts?

- Page 9 Line 1: Which version of the Copernicus DEM is used here? It would be more precise to use the abbreviation GLO-30 or GLO-90 for the 30m or 90m version respectively instead of "Copernicus DEM".

- Page 9 Line 12: While Hachmeister et al. (2022) mention that the Copernicus DEM is used in the updated version of the XCH4 data product, their analysis is based on comparisons between GMTED2010 and ICESat-2 data. Schneising et al. (2023) describe the updated XCH4 data. This should be clarified.

- Page 9 Line 18: I do not understand what is meant by the word "fidelity" in this context.

- Page 9 Line 23: Please specify "most regions".

- Page 9 Line 27: Please explain the abbreviation "TIN".

- Page 10 Line 7: What resolution is used your work?

- Page 11 Line 25: What is the motivation for this collocation criterion?

- Page 12 Table 3 & 4: It might be a good idea to combine these tables since they have the same column names.

- Page 13 Table 5: What is the reason for this selection of TCCON sites? It seems that only a small number of sites are left out from the analysis. It seems to me that especially Ny-Ålesund should be included in the analysis, since (a) the largest differences between DEMs lie in the high latitudes and (b) the topography around the Ny-Ålesund site is mountainous.

- Page 34 Figure 4: Why is there a gap in the middle of subfigure (b)? It should be explained where this is coming from.

---

## Author Comment (AC1)

Dear Reviewer #2 (RC1),

The authors would like to sincerely thank the reviewer for taking the time to evaluate this paper and provide useful, constructive feedback. We have made every effort to address the reviewer's comments and revise the paper accordingly. Detailed responses follow.

RC1: 'Comment on amt-2023-151', Anonymous Referee #2, 21 Sep 2023

Data description of OCO-2 retrievals

In the manuscript measurements taken in target, land nadir and land glint mode are mentioned and used. Please provide some explanation on the properties and differences between measurements taken in these modes. This would be especially important in context of the results presented in Table 6. Why was the initial bias in v10 lower in LNLG measurements compared to target mode measurements?

Response: While this detail is interesting and warrants further investigation, it is not relevant to the conclusions of this paper, which are centered around the impacts of the DEM on XCO2 bias. It is still the case that for both LNLG and target mode observations the overall mean bias and standard deviations in bias are reduced in v11.1 relative to either v10 or v11. This is the message we are aiming to send with these TCCON comparisons. Also note that although the mean biases are slightly larger for target mode retrievals, the standard deviations in bias are significantly reduced, likely because the observations are concentrated closer to the location of the TCCON site.

We could speculate on a few possible reasons for the reduced mean bias with LNLG. It may be related to the fact that the target mode measurements cover a larger range of viewing geometries than the nadir or glint measurements. It also may be due to the fact that LNLG measurements are more numerous and spread more evenly throughout the year. It is possible that for some sites, target mode observations can be more concentrated during certain seasons with more favorable weather or resources for coordinating target measurements. As a result, seasonal biases may skew the overall mean bias of the target mode observations.

Inversion results

In section 4.6 the impact of the DEM on flux estimates is discussed by comparing both the NASADEM+ and Copernicus DEM fluxes to the OCODEM fluxes. To me it seems that no significant difference in zonal mean fluxes are visible when using the Copernicus DEM. When using NASADEM+ the difference in zonal mean flux is however clearly visible. I think this is not sufficiently discussed in the paper and caused some confusion for me.

1. In the abstract and conclusion the stronger differences when using NASA-DEM+ are highlighted. In v11.1 the NASADEM+ is however replaced by the Copernicus DEM which shows no such differences. This seems confusing to me and this highlighting needs more justification (or it should be removed). At first glance this can be interpreted as an

exciting result gained from using a new DEM (compared to v10). But I suppose this is meant to discourage the usage of v11 in comparison to v11.1?

Response: Yes, we have revised the abstract to provide more context for these results, which likely show that NASADEM+ was an outlier, due to the anomalous mean altitude north of 60N. The flux inversion results further motivate the use of the Copernicus DEM, and acts as a warning to other groups in what can happen with a slightly erroneous DEM.

2. In Figure 13 panels (e) and (h) look very similar, while (f) and (i) (or (d) and (g)) do not. Why is this the case?

Response: It may be that regional patterns obscure the visibility of zonal mean structures. We tried to plot these results as clearly as possible by using a perceptually uniform colormap, and we have now reduced the range on the colorscale for (e) and (h) to make flux differences more visible. Note that, outside the high latitudes (which account for a relatively small fraction of total observations), panels (d) and (g) look quite similar.

3. In Figure 13 the maps are shown with a 4°x5° resolution, which is coarser than the resolution used when comparing DEMs. Why is this the case? Is this the resolution of the flux inversion? This should be clarified and also mentioned in section 3.4.

Response: We have revised the text as follows:

> Three sets of atmospheric CO2 inverse analyses were conducted for each of the original v10 (with OCODEM), NASADEM+5 modified, and Copernicus DEM modified XCO2 Land Nadir + Land Glint (LNLG) datasets. These inversions follow the set-up of Byrne et al. (2020**), and employ the CMS-Flux system with tracer transport at 4x5 degree spatial resolution using MERRA-2 reanalysis fields. We optimize 14-day scaling factors for each 4x5 degree grid cell on** net ecosystem exchange (NEE) and ocean surface-atmosphere fluxes for October 2017 through March 2019.  This is performed using three different prior NEE datasets, which are described in Byrne et al. (2020). As a result, a mini-ensemble of flux estimates is generated for each of the three XCO2 datasets, yielding a total of nine model runs. Posterior fluxes are examined for 2018 only. **See Sec. 3 of Byrne et al. (2020) for additional details on the inversion set-up.**

Specific comments:

P1, L10

RC1 comment: Why highlight the flux differences when using NASADEM+, when you decide on using the Copernicus DEM in the updated version (which does not show significant flux differences)?

Response: Other evidence presented in the paper and in cited literature suggests that the Copernicus DEM is more accurate than the NASADEM+, so it seems reasonable to conclude that the shifts in inferred fluxes with the NASADEM+ are also erroneous. This is illustrating one way in which the v11.1 product is an improvement on v11, and justifies the version update. Some revisions have been made to clarify this. As part of our response to the other reviewer, we have also clarified in Sect. 2.1 that v11 has already been made publicly available. We, therefore, believe it is important to emphasize the possibility for erroneous results using this already public dataset (i.e., OCO-2 v11).

P2, L24

RC1 comment: Could you shortly specify what kind of flaws?

Response: This paragraph of the introduction has been extensively revised with citations added to reinforce the claim that correlations between biases in estimated or retrieved surface pressure and biases in retrieved XCO2 are ubiquitous in satellite-based measurements and noted broadly within the current literature. In addition, some maps of the bias in retrieved surface pressures (dP) in each version of the OCO-2 ACOS algorithm have been added as supplemental material and referenced in the paper.

P2, L26

RC1 comment: Please specify what kind of effects the fixed surface pressure would have. Was the retrieval with a fixed surface pressure informed by an updated DEM tested?

Response: Although it would be interesting, elaborating on these tests within the paper would likely require at least an additional section, if not an entire separate paper. We believe this is not a useful addition because the impacts of the DEM would persist regardless of whether the surface pressure is fixed at the prior or retrieved and bias corrected in post-processing. The wording has been changed to remove mention of the fixed surface pressure analysis and, instead, we discuss the prevalence of other satellite-based retrieval algorithms that either fix surface pressure at the prior or retrieve it and then need to bias correct their final product due to a correlation between surface pressure bias and Xgas bias.

P3, L1

RC1 comment: It would be helpful to provide a brief summary of the results from Hachmeister et al. (2022).

Response: The second paragraph of the introduction has been revised to include more details on the results by Hachmeister et al. (2022) and how those results illustrate the relevance of the DEM for other trace gas measurements.

P3, L22

RC1 comment: Is this relevant for you comparisons between different DEMs for the XCO2 retrieval? I don't think you follow up on this later in the manuscript.

Response: It seems important to define and describe what a DEM is and its features before discussing the merits of different DEMs in the context of OCO-2 retrievals. Furthermore, gridding, smoothing, and void filling techniques used in the different DEMs, as well as how they have been changed in different iterations of the DEMs over time, are discussed throughout section 2. Therefore, we believe both this paragraph and the sentence on this line provide information relevant to the paper. We have, however, reworded the text from:

"A DEM goes beyond simply measuring surface elevations and uses various techniques of gridding, smoothing, and void filling to construct a full, continuous global map of surface elevations."

to:

"It is also worth noting that each DEM utilizes varying techniques for gridding, smoothing, and void filling in order to construct a full, continuous global map of surface elevations. Sections 2.3 through 2.7 provide some details for each DEM studied in this research."

P4, L31

RC1 comment: Why these two gases in particular? Would other trace gases be similarly affected?

Response: A 10-meter error is about 1 hPa which is about a 0.1% effect for any trace gas column. Most trace gas measurements do not care about such low systematic errors, because there is much more variability in the target gas. Due to their long atmospheric lifetime, CO2 and to a lesser extent CH4 have relatively low variability against their background, and thus O(0.1%) effects matter.

This is discussed more in the Conclusions, we have appended "due to the high precision and accuracy requirements of these gases" to the last sentence of the introduction for clarification.

We also added more details about the relevance to other gases in the second paragraph of the introduction.

P5, Table 1

RC1 comment: I am not sure whether this table is necessary since it contains no additional information.

Response: While the information in this table is somewhat redundant, we have decided to leave it in because it may serve as a convenient quick reference for future readers.

P6, L24 – 25

RC1 comment: To me it is not clear what motivates these formulas. Where does the 0.016 come from and what is meant by: "co2ratio bias and h2oratio bias represent the piecewise linear fits"?

Response: This appendix has been reworded for clarity in the revised manuscript. We believe this should make the equations clearer. Regarding the 0.016 number, we now write: "h2o_ratio_uncert_high_TCWV represents the asymptotic value of the uncertainty in h2o_ratio at high TCWV, and is set to 0.016."

P8, L1 – 2

RC1 comment: How was the averaging performed? What is the order of magnitude of these artifacts?

Response: OCO-2 retrievals have a reported sounding latitude and longitude coordinate that is approximately the center of the sounding footprint (~1.3 x 2.25 km). As described in Sect. 3.1, the sounding altitude is calculated as the average of all DEM elevations whose grid point coordinates fall within the boundaries of the sounding footprint. To obtain the OCODEM altitudes, all sounding altitudes with sounding coordinates that fall within a given 0.1 x 0.1 degree cell on a global grid are averaged to estimate the elevation of that grid cell in the original source DEM (OCODEM). While this method is somewhat convoluted, it should be reasonably accurate provided that there are sufficient numbers of OCO-2 soundings and the coverage is spatially consistent over the entire globe. However, OCO-2 coverage is not perfectly continuous or even spatially consistent across different regions, so these spatial inconsistencies will influence our mapping of the OCODEM by this approach. A map of the number of OCO-2 v10 soundings included in the aggregation process has been added to supplemental materials and shows the striated pattern in the southern hemisphere that is reflected as a striated pattern in elevation differences in Fig. 2 (now Fig. 1 in the revised version of the paper). This

map suggests that these elevation differences are artifacts of the aggregation process and not real differences between the OCODEM and the NASADEM+ or Copernicus GLO-90 DEM. Some additional details have also been added to Sect. 2.3 to elaborate.

P9, L1

RC1 comment: Which version of the Copernicus DEM is used here? It would be more precise to use the abbreviation GLO-30 or GLO-90 for the 30m or 90m version respectively instead of "Copernicus DEM".

Response: We use GLO-90 and the paper has been updated to refer to the "Copernicus GLO-90" instead of the "Copernicus DEM". In addition the following was added to the end of Sect. 2.5: The Copernicus global DEM has been produced as 30 m (~1 arcsecond) and 90 m (~3 arcsecond) resolution gridded products, referred to as GLO-30 and GLO-90, respectively. In this analysis and in the ACOS OCO-2 v11.1 update, the Copernicus GLO-90 is used.  This matches the resolution of the OCODEM and NASADEM+ products that are also considered in this study.

P9, L12

RC1 comment: While Hachmeister et al. (2022) mention that the Copernicus DEM is used in the updated version of the XCH4 data product, their analysis is based on comparisons between GMTED2010 and ICESat-2 data. Schneising et al. (2023) describe the updated XCH4 data. This should be clarified.

Response: Text has been updated to read as follows:

These reported metrics exclude Greenland and Antarctica due to complications in validation analyses over regions with permanent ice and snow; however, we show in Fig. 2 that the Copernicus DEM is in good agreement with the ArcticDEM (see Sect. 2.6) over Greenland and we show in Fig. 3 that the Copernicus DEM is in good agreement with REMA (see Sect. 2.7) over Antarctica. Both the ArcticDEM and REMA are validated using ICESat-2, which is also shown by Hachmeister et al. (2022) to improve XCH4 retrievals from TROPOMI over Greenland. The findings of Hachmeister et al. (2022) prompted a change to the Copernicus GLO-90 DEM in the most recent update to TROPOMI XCH4 retrievals. As a result, Schneising et al. (2023) report reduced errors in assumed surface pressure and retrieved XCH4 on the order of 1\%, with notable improvements over high latitude regions.

P9, L18

RC1 comment: I do not understand what is meant by the word "fidelity" in this context.

Response: Changed to "accuracy and precision".

P9 L23

RC1 comment: Please specify "most regions".

Response: There are some voids in this DEM. The text has been revised to clarify.

P9, L27

RC1 comment: Please explain the abbreviation "TIN".

Response: Triangular Irregular Network (TIN). Definition added to the paper. Please refer to the cited literature for further details.

P10, L7

RC1 comment: What resolution is used your work?

Response: 32 m, the same as the ArcticDEM. This detail has been added.

P11, L25

RC1 comment: What is the motivation for this collocation criterion?

Response: The coincidence criteria described in the paper are standard criteria used by many OCO-2 validation and data comparison analyses. They are similar to the criteria used in the OCO-2 validation analysis by Wunch et al. 2017, and this paper probably set the precedent for other studies to use similar criteria. However, there is no justification provided by Wunch et al. for their choice of coincidence criteria, and it was likely a somewhat arbitrary choice in which they sought to balance the amount of data available for comparisons with the possibility of introducing colocation errors. There is actually a mistake in Sect. 3.3, the geographic criteria are slightly more restrictive in our analysis (5 x 5 degrees, rather than 5 x10 degrees). This has been corrected in the paper.

Taylor, AMT, 2023 Section 6.1 discusses more stringent collocation criteria of 2.5x5 degree single overpass means from OCO-2 versus the mean TCCON +/- 1hr of the OCO overpass time. In general, a lot more time has elapsed, allowing many more observations and the opportunity to be more conservative with coincidence criteria.

P12, Table 3 & 4

RC1 comment: It might be a good idea to combine these tables since they have the same column names.

Response: It seems useful to differentiate between variables that exist in v11, but have changed in v11.1, and variables that are new in v11.1, but do not exist in v11.

P13, Table 5

RC1 comment: What is the reason for this selection of TCCON sites? It seems that only a small number of sites are left out from the analysis. It seems to me that especially NyÅlesund should be included in the analysis, since (a) the largest differences between DEMs lie in the high latitudes and (b) the topography around the NyÅlesund site is mountainous.

Response: The TCCON sites that are excluded are those that do not have a sufficient quantity of coincident OCO-2 observations that pass quality controls. We have added the following details to Sect. 3.3 to explain the thresholds for inclusion of coincident pairings:

"Only OCO overpasses that have at least 100 good quality sounding and at least 10 TCCON measurements within the 2 hour period around the mean overpass time are included. "

This is likely the reason why NyÅlesund data are excluded. Other reasons include:
- We stopped taking targets over NyÅlesund early in the mission because of low SNR in the winter months. Also, soundings over snow surfaces are often considered as low quality.
- NyÅlesund is located on an island, so many OCO-2 soundings in the coincidence region are over ocean and this analysis only considers OCO-2 observations over land.

P34, Figure 4

RC1 comment: Why is there a gap in the middle of subfigure (b)? It should be explained where this is coming from.

Response: There are some tiles with reference coordinates that do not convert easily to geospatial coordinates or overlap extensively with neighboring mosaic tiles of the REMA DEM. Rather than erroneously represent these data they were left out. In addition, REMA has a void directly over the geographic south pole. The comparisons over Antarctica are not essential in the context of OCO-2 retrievals because the observational coverage is severely limited there. However, we still believe it is useful to present some of the results for Antarctica, as these may be of interest to other satellite missions.

---

## Author Comment (AC2)

Dear Reviewer #1 (RC2),

The authors would like to sincerely thank the reviewer for taking the time to evaluate this paper and provide useful, constructive feedback. We have made every effort to address the reviewer's comments and revise the paper accordingly. Detailed responses follow.

**RC2**: ['Comment on amt-2023-151'](), Anonymous Referee #1, 21 Sep 2023

> In this paper, Jacobs et al. evaluate the impact of different digital elevation models (DEMs) on XCO2 retrievals from the OCO-2 Instrument. By comparing different DEMs, including OCODEM used in the ACOS V10 algorithm, the NASADEM+ used in ACOS V11, and the Copernicus DEM used in ACOS V11.1, the authors conclude that the Copernicus DEM has better overall continuity and accuracy. The authors demonstrate that the differences in DEM have a significant impact on the bias-corrected XCO2 retrievals. With the use of the Copernicus DEM and updated quality control filtering, the ACOS V11.1 XCO2 retrievals show generally similar or improved accuracy and spatial continuity as compared with V10 and V11 retrievals. In addition, V11.1 also has increased the data volume that passes through the data quality filtering. This paper discusses an important input dataset for satellite trace gas retrievals. The results and recommendation from this study should be useful for both algorithm developers and data users. The paper is well organized, and the figures are mostly clear. However, I feel that in its current form, the paper is more tailored towards readers who are already familiar with (or have working knowledge in) the ACOS retrieval algorithm. It would be fine as a chapter within the ATBD or product readme file, as in those documents, the other chapters would have provided the necessary context. As a stand-alone paper, I found it rather difficult to follow. Some terms and parameters are casually introduced with little explanation. Some important conclusions are drawn without the necessary supporting evidence.

Response: Some additional details have been added to Sect. 2.1 and throughout the text. As before, we cite the OCO-2 Algorithm Theoretical Basis Document (ATBD) and several other relevant publications in Sect. 2.1 of the paper, which should point readers who are less familiar with OCO-2 operations and retrievals to useful and informative resources. For a full explanation of how the OCO-2 instrument and retrieval operate, the ATBD and other cited literature , it is not practical for us to reiterate everything in this paper. The most recent OCO-2 ATBD can be accessed at [https://docserver.gesdisc.eosdis.nasa.gov/public/project/OCO/OCO_L2_ATBD.pdf](https://docserver.gesdisc.eosdis.nasa.gov/public/project/OCO/OCO_L2_ATBD.pdf). It should be sufficient to provide citations to literature that already discuss background information, allowing readers who need more details to seek it out within those cited works. This paper does present results that are relevant to other satellite missions and retrievals. Nearly all satellite-based retrievals of trace gases struggle with the accuracy of their surface pressure retrievals, or do not retrieve surface pressure at all, and in most cases a DEM is required to rescale surface pressure from a meteorological model or reanalysis product. Furthermore, even the meteorological models and reanalysis products used for prior surface

pressure estimates use some source of DEM data. It is important for many groups aside from only users of the OCO-2 product to consider the impacts of a poor choice of DEM on their data products, which should justify this paper as its own publication and not simply a chapter in the OCO-2 ATBD. In addition, to align more with terminology in Kiel et al. (2019), we have changed "dpfrac" to $dP_{frac}$ be more in line with his definitions. Regarding the latter statement, It is unclear to which conclusions the reviewer is referring. We would need more specificity from the reviewer in order to properly address their concern.

Specific comments:

P2, L6

RC2 comment: can you provide a reference that the 14-year GOSAT data record is long enough to inform multi-decadal climate variations?

Response: Changed to "accumulated data records long enough to describe interannual climate variations and characterize seasonal cycles" and several citations are added.

P2, L25

RC2 comment: can the authors elaborate more on the tests that fix surface pressures? What are the other deleterious effects and why? This would be interesting, given that there are GHG retrieval algorithms that use assimilated surface pressures (rather than retrieving them).

Response: Although it would be interesting, elaborating on these tests within the paper would likely require at least an additional section, if not an entirely separate paper. We believe this is not a useful addition because the impacts of the DEM would persist regardless of whether the surface pressure is fixed at the prior or retrieved and bias corrected in post-processing. The wording has been changed to remove mention of the fixed surface pressure analysis and, instead, we discuss the prevalence of other satellite-based retrieval algorithms that either fix surface pressure at the prior or retrieve it and then have to bias correct their final product due to a correlation between surface pressure bias and Xgas bias.

P5, L5

RC2 comment: this part repeats what has already been introduced in page 2.

Response: This has been updated to read as follows:

The OCO-2 instrument uses observed spectral radiances in the O2A band to retrieve estimates of surface pressure necessary for retrieving XCO2 (see Sect. 2.1 and 3.2).

Section 2.1

RC2 comment: can the authors clarify on the status of OCO-2 XCO2 products? Will V11 be released or only V11.1 be released?

Response: Both v11 and v11.1 have been released and are currently available on GES-DISC, but the default OCO-2 $XCO_2$ product is now v11.1, which we encourage users to use. This has been made clear in the paper.

P5, L25

RC2 comment: can you briefly explain the footprint bias correction (it is also in equation 6)?

Response: The multiplicative scaling is to ensure no global bias with respect to TCCON.  The footprint correction is an additive offset of a few tenths of a ppm or less to ensure there is no mean difference amongst OCO-2's eight footprints. There are two citations provided in the paper that describe the footprint bias corrections in detail, as well as all other terms in the bias correction and how they are calculated. The details of the bias correction that are not thoroughly explained in the paper are left out because they are not particularly relevant to the analysis presented in this paper. See also Kiel et al. (2019).

Section 2.1.1

RC2 comment: could you please rewrite this section to help those who are less familiar with ACOS to better understand it?

Response: This section was moved to the appendix because it is incidental to the  primary focus of this study, which is the digital elevation map and its impact on XCO2 and subsequent inferred CO2 fluxes.  We only include the changes from v10 to v11 as background information for interested readers, which motivated the move to an appendix.

Equation 6

RC2 comment: where does "0.016" come from? One can guess from the figure, but it would be helpful if the authors can give some explanation.

Response: This appendix has been reworded for clarity in the revised manuscript.  In particular, we now write:

"h2o_ratio_uncert_high_TCWV represents the asymptotic value of the uncertainty in h2o_ratio at high TCWV and is set to 0.016."

P6, L28

RC2 comment: can the authors give some quantitative results on how much more robust the new filtering is?

Response: This is shown in Fig. 9 and discussed in Sect. 4.4. We have added a sentence to Sect. 2.1 pointing to this.

P8, L1

RC2 comment: is the aggregation done to all DEMs or just OCODEM?

Response: Aggregations were done on all DEMs for the purposes of comparison, as described in Sect. 3.1, but only the OCODEM altitude were extracted from OCO-2 sounding retrievals. We were not able to obtain access to the original DEM data from the OCODEM, and therefore the OCODEM elevations have already been averaged over the sounding field of view before being aggregated. It also means that we only have OCODEM elevations for regions where there are OCO-2 soundings that pass preliminary screening and have a valid retrieval. We have revised the paper to clarify this.

P9, L13

RC2 comment: GMTED2010 is not defined or used elsewhere in the paper.

Response: Global multi-resolution terrain elevation data 2010 (GMTED2010) is the DEM used in TROPOMI retrievals prior to the most recent update to the Copernicus DEM. Mention of this DEM has been removed from the paper and its analysis is not relevant to this study. We have also made revisions in regard to our discussion of results from Hachmeister et al. (2022) and the DEM update in TROPOMI retrievals in response to comments by RC1.

Section 3.1

RC2 comment: it would be good to run V11 full physics algorithm for a small subset of OCO-2 data to confirm that you will get the same bias-corrected XCO2 as V11.1.

Response: Due to computational expense and lack of available human resources at JPL, it is not feasible to run the v11 L2FP on a subset of the data with the Copernicus DEM.

Section 4.5

RC2 comment: it could be useful to produce maps for Lauder, Pasadena, and Eureka that are similar to Figure 11 (even if just place them in the supplemental information).

Response: Some additional details on the target measurements at these three sites have been added to the supplemental materials and the relevant section of the supplement is referenced in the paper.

Figure 1

RC2 comment: fitting at low TCWV range appears to be of worse quality - how does this affect the corrected h2o_ratio and the overall results, given that significant improvement is seen over low TCWV?

Response: It's relative to "no correction at all". Clearly at low TCWV, there is a bias in h2o_ratio; even if there is uncertainty in the correction, it is much better than no correction. Plus, it is likely that other factors may also affect the clear-sky bias in h2o_ratio, such as surface albedo, signal-to-noise ratios, etc. But this is the largest such factor. Also, note that this figure has been moved to an appendix along with the relevant section, and is no longer Fig. 1 in the revised version of the paper.

Figure 5

RC2 comment: consider putting mean and standard deviation of delta-altitude in different panels so that the differences in the mean between NASADEM+ and Copernicus DEM are more obvious.

Response: This figure has been updated as suggested.

---

## Author Response (AR2)

Second response to Referee #2 comments:

- Line 11 (and all other occurrences): I'd suggest to replace "Copernicus GLO-90" with "Copernicus GLO-90 DEM", and after the first mention by "GLO-90 DEM" to help with readability

Response: changed as suggested.

- Line 77, 310: You should add mention that this refers to the WFMD retrieval algorithm and not the operational retrieval algorithm to avoid confusion. I would recommend replacing "TROPOMI retrieval algorithm" with "TROPOMI WFMD retrieval algorithm" everywhere in the manuscript

Response: changed as suggested.

- Line 161, 355: Replace km with km²

Response: changed as suggested.

- Line 380: It would be helpful to add a citation to this claim ([...] reanalyses are believed to be better than 1 hPa.)

Response: changed to read "…reanalyses are believed to be better than 1 hPa, though more work is needed to evaluate this." Apologies, but we were not able to find a recent, robust, and global study of uncertainties and errors in surface pressures from meteorological reanalyses.